# Entropy Rate Estimation for Markov Chains with Large State Space

**Yanjun Han**
Department of Electrical Engineering
Stanford University
Stanford, CA 94305
yjhan@stanford.edu

**Jiantao Jiao**
Department of Electrical Engineering and Computer Sciences
University of California, Berkeley
Berkeley, CA 94720
jiantao@berkeley.edu

**Chuan-Zheng Lee, Tsachy Weissman**
Department of Electrical Engineering
Stanford University
Stanford, CA 94305
{czlee, tsachy}@stanford.edu

**Yihong Wu**
Department of Statistics and Data Science
Yale University
New Haven, CT 06511
yihong.wu@yale.edu

**Tiancheng Yu**
Department of Electronic Engineering
Tsinghua University
Haidian, Beijing 100084
thueeyutc14@foxmail.com

## Abstract

Entropy estimation is one of the prototypical problems in distribution property testing. To consistently estimate the Shannon entropy of a distribution on $S$ elements with independent samples, the optimal sample complexity scales sublinearly with $S$ as $\Theta(\frac{S}{\log S})$ as shown by Valiant and Valiant [4]. Extending the theory and algorithms for entropy estimation to dependent data, this paper considers the problem of estimating the entropy rate of a stationary reversible Markov chain with $S$ states from a sample path of $n$ observations. We show that

- Provided the Markov chain mixes not too slowly, *i.e.*, the relaxation time is at most $O(\frac{S}{\ln^3 S})$, consistent estimation is achievable when $n \gg \frac{S^2}{\log S}$.

- Provided the Markov chain has some slight dependency, *i.e.*, the relaxation time is at least $1 + \Omega(\frac{\ln^2 S}{\sqrt{S}})$, consistent estimation is impossible when $n \lesssim \frac{S^2}{\log S}$.

Under both assumptions, the optimal estimation accuracy is shown to be $\Theta(\frac{S^2}{n \log S})$. In comparison, the empirical entropy rate requires at least $\Omega(S^2)$ samples to be consistent, even when the Markov chain is memoryless. In addition to synthetic experiments, we also apply the estimators that achieve the optimal sample complex-

ity to estimate the entropy rate of the English language in the Penn Treebank and the Google One Billion Words corpora, which provides a natural benchmark for language modeling and relates it directly to the widely used perplexity measure.

# 1   Introduction

Consider a stationary stochastic process $\{X_t\}_{t=1}^{\infty}$, where each $X_t$ takes values in a finite alphabet $\mathcal{X}$ of size $S$. The *Shannon entropy rate* (or simply *entropy rate*) of this process is defined as [1]

$$\bar{H} = \lim_{n \to \infty} \frac{1}{n} H(X^n), \tag{1}$$

where

$$H(X^n) = \sum_{x^n \in \mathcal{X}^n} P_{X^n}(x^n) \ln \frac{1}{P_{X^n}(x^n)}$$

is the *Shannon entropy* (or *entropy*) of the random vector $X^n = (X_1, X_2, \ldots, X_n)$ and $P_{X^n}(x^n) = \mathbb{P}(X_1 = x_1, \ldots, X_n = x_n)$ is the joint probability mass function. Since the entropy of a random variable depends only on its distribution, we also refer to the entropy $H(P)$ of a discrete distribution $P = (p_1, p_2, \ldots, p_S)$, defined as $H(P) = \sum_{i=1}^{S} p_i \ln \frac{1}{p_i}$.

The Shannon entropy rate is the fundamental limit of the expected logarithmic loss when predicting the next symbol, given the all past symbols. It is also the fundamental limit of data compressing for stationary stochastic processes in terms of the average number of bits required to represent each symbol [1, 7]. Estimating the entropy rate of a stochastic process is a fundamental problem in information theory, statistics, and machine learning; and it has diverse applications—see, for example, [3, 2, 3, 3, 4, 2].

There exists extensive literature on entropy rate estimation. It is known from data compression theory that the normalized codelength of any *universal* code is a consistent estimator for the entropy rate as the number of samples approaches infinity. This observation has inspired a large variety of entropy rate estimators; see *e.g.* [4, 2, 1, 6, 1]. However, most of this work has been in the asymptotic regime [3, 8]. Attention to *non-asymptotic* analysis has only been more recent, and to date, almost only for i.i.d. data. There has been little work on the non-asymptotic performance of an entropy rate estimator for dependent data—that is, where the alphabet size is large (making asymptotically large datasets infeasible) and the stochastic process has memory. An understanding of this large-alphabet regime is increasingly important in modern machine learning applications, in particular, *language modeling*. There have been substantial recent advances in probabilistic language models, which have been widely used in applications such as machine translation and search query completion. The entropy rate of (say) the English language represents a fundamental limit on the efficacy of a language model (measured by its *perplexity*), so it is of great interest to language model researchers to obtain an accurate estimate of the entropy rate as a benchmark. However, since the alphabet size here is exceedingly large, and Google's One Billion Words corpus includes about two million unique words,[1] it is unrealistic to assume the large-sample asymptotics especially when dealing with combinations of words (bigrams, trigrams, etc). It is therefore of significant practical importance to investigate the optimal entropy rate estimator with limited sample size.

In the context of non-asymptotic analysis for i.i.d. samples, Paninski [3] first showed that the Shannon entropy can be consistently estimated with $o(S)$ samples when the alphabet size $S$ approaches infinity. The seminal work of [4] showed that when estimating the entropy rate of an i.i.d. source, $n \gg \frac{S}{\log S}$ samples are necessary and sufficient for consistency. The entropy estimators proposed in [4] and refined in [4], based on linear programming, have not been shown to achieve the minimax estimation rate. Another estimator proposed by the same authors [4] has been shown to achieve the minimax rate in the restrictive regime of $\frac{S}{\ln S} \lesssim n \lesssim \frac{S^{1.03}}{\ln S}$. Using the idea of best polynomial approximation, the independent work of [4] and [1] obtained estimators that achieve the minimax mean-square error $\Theta((\frac{S}{n \log S})^2 + \frac{\log^2 S}{n})$ for entropy estimation. The intuition for the $\Theta(\frac{S}{\log S})$ sample complexity in the

independent case can be interpreted as follows: as opposed to estimating the entire distribution which has $S - 1$ parameters and requires $\Theta(S)$ samples, estimating the scalar functional (entropy) can be done with a logarithmic factor reduction of samples. For Markov chains which are characterized by the transition matrix consisting of $S(S-1)$ free parameters, it is reasonable to expect an $\Theta(\frac{S^2}{\log S})$ sample complexity. Indeed, we will show that this is correct provided the mixing is not too slow.

Estimating the entropy rate of a Markov chain falls in the general area of property testing and estimation with dependent data. The prior work [2] provided a non-asymptotic analysis of maximum-likelihood estimation of entropy rate in Markov chains and showed that it is necessary to assume certain assumptions on the mixing time for otherwise the entropy rate is impossible to estimate. There has been some progress in related questions of estimating the mixing time from sample path [1, 2], estimating the transition matrix [1], and testing symmetric Markov chains [1]. The current paper makes contribution to this growing field. In particular, the main results of this paper are highlighted as follows:

- We provide a tight analysis of the sample complexity of the empirical entropy rate for Markov chains when the mixing time is not too large. This refines results in [2] and shows that when mixing is not too slow, the sample complexity of the empirical entropy does not depend on the mixing time. Precisely, the bias of the empirical entropy rate vanishes uniformly over all Markov chains regardless of mixing time and reversibility as long as the number of samples grows faster than the number of parameters. It is its variance that may explode when the mixing time becomes gigantic.

- We obtain a characterization of the optimal sample complexity for estimating the entropy rate of a stationary reversible Markov chain in terms of the sample size, state space size, and mixing time, and partially resolve one of the open questions raised in [2]. In particular, we show that when the mixing is neither too fast nor too slow, the sample complexity (up to a constant) does not depend on mixing time. In this regime, the performance of the optimal estimator with $n$ samples is essentially that of the empirical entropy rate with $n \log n$ samples. As opposed to the lower bound for estimating the mixing time in [1] obtained by applying Le Cam's method to two Markov chains which are statistically indistinguishable, the minimax lower bound in the current paper is much more involved, which, in addition to a series of reductions by means of simulation, relies on constructing two stationary reversible Markov chains with *random* transition matrices [4], so that the marginal distributions of the sample paths are statistically indistinguishable.

- We construct estimators that are efficiently computable and achieve the minimax sample complexity. The key step is to connect the entropy rate estimation problem to Shannon entropy estimation on large alphabets with i.i.d. samples. The analysis uses the idea of simulating Markov chains from independent samples by Billingsley [3] and concentration inequalities for Markov chains.

- We compare the empirical performance of various estimators for entropy rate on a variety of synthetic data sets, and demonstrate the superior performances of the information-theoretically optimal estimators compared to the empirical entropy rate.

- We apply the information-theoretically optimal estimators to estimate the entropy rate of the Penn Treebank (PTB) and the Google One Billion Words (1BW) datasets. We show that even only with estimates using up to 4-grams, there may exist language models that achieve better perplexity than the current state-of-the-art.

The rest of the paper is organized as follows. After setting up preliminary definitions in Section 2, we summarize our main findings in Section 3, with proofs sketched in Section 4. Section 5 provides empirical results on estimating the entropy rate of the Penn Treebank (PTB) and the Google One Billion Words (1BW) datasets. Detailed proofs and more experiments are deferred to the appendices.

## 2   Preliminaries

Consider a first-order Markov chain $X_0, X_1, X_2, \ldots$ on a finite state space $\mathcal{X} = [S]$ with transition kernel $T$. We denote the entries of $T$ as $T_{ij}$, that is, $T_{ij} = P_{X_2|X_1}(j|i)$ for $i, j \in \mathcal{X}$. Let $T_i$ denote the $i$th row of $T$, which is the conditional law of $X_2$ given $X_1 = i$. Throughout the paper, we focus

on first-order Markov chains, since any finite-order Markov chain can be converted to a first-order one by extending the state space [3].

We say that a Markov chain is *stationary* if the distribution of $X_1$, denoted by $\pi \triangleq P_{X_1}$, satisfies $\pi T = \pi$. We say that a Markov chain is *reversible* if it satisfies the detailed balance equations, $\pi_i T_{ij} = \pi_j T_{ji}$ for all $i, j \in \mathcal{X}$. If a Markov chain is reversible, the (left) spectrum of its transition matrix $T$ contains $S$ real eigenvalues, which we denote as $1 = \lambda_1 \geq \lambda_2 \geq \cdots \geq \lambda_S \geq -1$. We define the *spectral gap* and the *absolute spectral gap* of $T$ as $\gamma(T) = 1 - \lambda_2$ and $\gamma^*(T) = 1 - \max_{i \geq 2} |\lambda_i|$, respectively, and the *relaxation time* of a reversible Markov chain as

$$\tau_{\text{rel}}(T) = \frac{1}{\gamma^*(T)}. \tag{2}$$

The relaxation time of a reversible Markov chain (approximately) captures its mixing time, which roughly speaking is the smallest $n$ for which the marginal distribution of $X_n$ is close to the Markov chain's stationary distribution. We refer to [3] for a survey.

We consider the following observation model. We observe a sample path of a stationary finite-state Markov chain $X_0, X_1, \ldots, X_n$, whose Shannon entropy rate $\bar{H}$ in (1) reduces to

$$\bar{H} = \sum_{i=1}^{S} \pi_i \sum_{j=1}^{S} T_{ij} \ln \frac{1}{T_{ij}} = H(X_1, X_2) - H(X_1) \tag{3}$$

where $\pi$ is the stationary distribution of this Markov chain. Let $\mathcal{M}_2(S)$ be the set of transition matrices of all stationary Markov chains on a state space of size $S$. Let $\mathcal{M}_{2,\text{rev}}(S)$ be the set of transition matrices of all stationary *reversible* Markov chains on a state space of size $S$. We define the following class of stationary Markov reversible chains whose relaxation time is at most $\frac{1}{\gamma^*}$:

$$\mathcal{M}_{2,\text{rev}}(S, \gamma^*) = \{T \in \mathcal{M}_{2,\text{rev}}(S), \gamma^*(T) \geq \gamma^*\}. \tag{4}$$

The goal is to characterize the sample complexity of entropy rate estimation as a function of $S, \gamma^*$, and the estimation accuracy.

Note that the entropy rate of a first-order Markov chain can be written as

$$\bar{H} = \sum_{i=1}^{S} \pi_i H(X_2 | X_1 = i). \tag{5}$$

Given a sample path $\mathbf{X} = (X_0, X_1, \ldots, X_n)$, let $\hat{\pi}$ denote the empirical distribution of states, and the subsequence of $\mathbf{X}$ containing elements *following* any occurrence of the state $i$ as $\mathbf{X}^{(i)} = \{X_j : X_j \in \mathbf{X}, X_{j-1} = i, j \in [n]\}$. A natural idea to estimate the entropy rate $\bar{H}$ is to use $\hat{\pi}_i$ to estimate $\pi_i$ and an appropriate Shannon entropy estimator to estimate $H(X_2 | X_1 = i)$. We define two estimators:

1. The *empirical entropy rate*: $\bar{H}_{\text{emp}} = \sum_{i=1}^{S} \hat{\pi}_i \hat{H}_{\text{emp}}(\mathbf{X}^{(i)})$. Note that $\hat{H}_{\text{emp}}(\mathbf{Y})$ computes the Shannon entropy of the empirical distribution of its argument $\mathbf{Y} = (Y_1, Y_2, \ldots, Y_m)$.

2. Our entropy rate estimator: $\bar{H}_{\text{opt}} = \sum_{i=1}^{S} \hat{\pi}_i \hat{H}_{\text{opt}}(\mathbf{X}^{(i)})$, where $\hat{H}_{\text{opt}}$ is any minimax rate-optimal Shannon entropy estimator designed for i.i.d. data, such as those in [4, 4, 1].

## 3 Main results

Our first result provides performance guarantees for the empirical entropy rate $\bar{H}_{\text{emp}}$ and our entropy rate estimator $\bar{H}_{\text{opt}}$:

**Theorem 1.** *Suppose $(X_0, X_1, \ldots, X_n)$ is a sample path from a stationary reversible Markov chain with spectral gap $\gamma$. If $S^{0.01} \lesssim n \lesssim S^{2.99}$ and $\frac{1}{\gamma} \lesssim \frac{S}{\ln n \ln^2 S} \wedge \frac{S^3}{n \ln n \ln^3 S}$, there exists some constant $C > 0$ independent of $n, S, \gamma$ such that the entropy rate estimator $\bar{H}_{\text{opt}}$ satisfies:[2] as $S \to \infty$,*

$$P\left(|\bar{H}_{\text{opt}} - \bar{H}| \leq C \frac{S^2}{n \ln S}\right) \to 1 \tag{6}$$

*Under the same conditions, there exists some constant $C' > 0$ independent of $n, S, \gamma$ such that the empirical entropy rate $\bar{H}_{\mathsf{emp}}$ satisfies: as $S \to \infty$,*

$$P\left(|\bar{H}_{\mathsf{emp}} - \bar{H}| \leq C'\frac{S^2}{n}\right) \to 1. \tag{7}$$

Theorem 1 shows that when the sample size is not too large, and the mixing is not too slow, it suffices to take $n \gg \frac{S^2}{\ln S}$ for the estimator $\bar{H}_{\mathsf{opt}}$ to achieve a vanishing error, and $n \gg S^2$ for the empirical entropy rate. Theorem 1 improves over [2] in the analysis of the empirical entropy rate in the sense that unlike the error term $O(\frac{S^2}{n\gamma})$, our dominating term $O(\frac{S^2}{n})$ does not depend on the mixing time.

Note that we have made mixing time assumptions in the upper bound analysis of the empirical entropy rate in Theorem 1, which is natural since [2] showed that it is necessary to impose mixing time assumptions to provide meaningful statistical guarantees for entropy rate estimation in Markov chains. The following result shows that mixing assumptions are only needed to control the variance of the empirical entropy rate: the bias of the empirical entropy rate vanishes uniformly over all Markov chains regardless of reversibility and mixing time assumptions as long as $n \gg S^2$.

**Theorem 2.** *Let $n, S \geq 1$. Then,*

$$\sup_{T \in \mathcal{M}_2(S)} |\bar{H} - \mathbb{E}[\bar{H}_{\mathsf{emp}}]| \leq \frac{2S^2}{n}\ln\left(\frac{n}{S^2} + 1\right) + \frac{(S^2 + 2)\ln 2}{n}. \tag{8}$$

Theorem 2 implies that if $n \gg S^2$, the bias of the empirical entropy rate estimator universally vanishes for any stationary Markov chains.

Now we turn to the lower bounds, which show that the scalings in Theorem 1 are in fact tight. The next result shows that the bias of the empirical entropy rate $\bar{H}_{\mathsf{emp}}$ is non-vanishing unless $n \gg S^2$, even when the data are independent.

**Theorem 3.** *If $\{X_0, X_1, \ldots, X_n\}$ are mutually independent and uniformly distributed, then*

$$|\bar{H} - \mathbb{E}[\bar{H}_{\mathsf{emp}}]| \geq \ln\left(\frac{S^2}{n + S - 1}\right). \tag{9}$$

The following corollary is immediate.

**Corollary 1.** *There exists a universal constant $c > 0$ such that when $n \leq cS^2$, the absolute value of the bias of $\bar{H}_{\mathsf{emp}}$ is bounded away from zero even if the Markov chain is memoryless.*

The next theorem presents a minimax lower bound for entropy rate estimation which applies to any estimation scheme regardless of its computational cost. In particular, it shows that $\bar{H}_{\mathsf{opt}}$ is minimax rate-optimal under mild assumptions on the mixing time.

**Theorem 4.** *For $n \geq \frac{S^2}{\ln S}, \ln n \ll \frac{S}{(\ln S)^2}, \gamma^* \leq 1 - C_2\sqrt{\frac{S\ln^3 S}{n}}$, we have*

$$\liminf_{S \to \infty} \inf_{\hat{H}} \sup_{T \in \mathcal{M}_{2,rev}(S,\gamma^*)} P\left(|\hat{H} - \bar{H}| \geq C_1\frac{S^2}{n\ln S}\right) \geq \frac{1}{2}. \tag{10}$$

*Here $C_1, C_2$ are universal constants from Theorem 6.*

The following corollary, which follows from Theorem 1 and 4, presents the critical scaling that determines whether consistent estimation of the entropy rate is possible.

**Corollary 2.** *If $\frac{\ln^3 S}{S} \ll \gamma^* \leq 1 - C_2\frac{\ln^2 S}{\sqrt{S}}$, there exists an estimator $\hat{H}$ which estimates the entropy rate with a uniformly vanishing error over Markov chains $\mathcal{M}_{2,rev}(S,\gamma^*)$ if and only if $n \gg \frac{S^2}{\ln S}$.*

To conclude this section we summarize our result in terms of the sample complexity for estimating the entropy rate within a few bits ($\epsilon = \Theta(1)$), classified according to the relaxation time:

- $\tau_{\mathrm{rel}} = 1$: this is the i.i.d. case and the sample complexity is $\Theta(\frac{S}{\ln S})$;

- $1 < \tau_{\text{rel}} \ll 1 + \Omega(\frac{\ln^2 S}{\sqrt{S}})$: in this narrow regime the sample complexity is at most $O(\frac{S^2}{\ln S})$ and no matching lower bound is known;

- $1 + \Omega(\frac{\ln^2 S}{\sqrt{S}}) \leq \tau_{\text{rel}} \ll \frac{S}{\ln^3 S}$: the sample complexity is $\Theta(\frac{S^2}{\ln S})$;

- $\tau_{\text{rel}} \gtrsim \frac{S}{\ln^3 S}$: the sample complexity is $\Omega(\frac{S^2}{\ln S})$ and no matching upper bound is known. In this case the chain mixes very slowly and it is likely that the variance will dominate.

## 4 Sketch of the proof

In this section we sketch the proof of Theorems 1, 2 and 4, and defer the details to the appendix.

### 4.1 Proof of Theorem 1

A key step in the analysis of $\bar{H}_{\text{emp}}$ and $\bar{H}_{\text{opt}}$ is the idea of simulating a finite-state Markov chain from independent samples [3, p. 19]: consider an independent collection of random variables $X_0$ and $W_{in}$ $(i = 1, 2, \ldots, S; n = 1, 2, \ldots)$ such that $P_{X_0}(i) = \pi_i, P_{W_{in}}(j) = T_{ij}$. Imagine the variables $W_{in}$ set out in the following array:

$$
\begin{array}{ccccc}
W_{11} & W_{12} & \ldots & W_{1n} & \ldots \\
W_{21} & W_{22} & \ldots & W_{2n} & \ldots \\
\vdots & \vdots & \ddots & \vdots & \vdots \\
W_{S1} & W_{S2} & \ldots & W_{Sn} & \ldots
\end{array}
$$

First, $X_0$ is sampled. If $X_0 = i$, then the first variable in the $i$th row of the array is sampled, and the result is assigned by definition to $X_1$. If $X_1 = j$, then the first variable in the $j$th row is sampled, unless $j = i$, in which case the second variable is sampled. In any case, the result of the sampling is by definition $X_2$. The next variable sampled is the first one in row $X_2$ which has not yet been sampled. This process thus continues. After collecting $\{X_0, X_1, \ldots, X_n\}$ from the model, we assume that the last variable sampled from row $i$ is $W_{in_i}$. It can be shown that observing a Markov chain $\{X_0, X_1, \ldots, X_n\}$ is equivalent to observing $\{X_0, \{W_{ij}\}_{i \in [S], j \in [n_i]}\}$, and consequently $\hat{\pi}_i = n_i/n, \mathbf{X}^{(i)} = (W_{i1}, \ldots, W_{in_i})$.

The main reason to introduce the above framework is to analyze $\hat{H}_{\text{emp}}(\mathbf{X}^{(i)})$ and $\hat{H}_{\text{opt}}(\mathbf{X}^{(i)})$ as if the argument $\mathbf{X}^{(i)}$ is an i.i.d. vector. Specifically, although $W_{i1}, \cdots, W_{im}$ conditioned on $n_i = m$ are not i.i.d., they are i.i.d. as $T_i$ for any *fixed* $m$. Hence, using the fact that each $n_i$ concentrates around $n\pi_i$ (cf. Definition 2 and Lemma 4 for details), we may use the concentration properties of $\hat{H}_{\text{emp}}$ and $\hat{H}_{\text{opt}}$ (cf. Lemma 3) on i.i.d. data for each *fixed* $m \approx n\pi_i$ and apply the union bound in the end.

Based on this alternative view, we have the following theorem, which implies Theorem 1.

**Theorem 5.** *Suppose* $(X_0, X_1, \ldots, X_n)$ *comes from a stationary reversible Markov chain with spectral gap* $\gamma$. *Then, with probability tending to one, the entropy rate estimators satisfy*

$$
|\bar{H}_{\text{opt}} - \bar{H}| \lesssim \frac{S^2}{n \ln S} + \left(\frac{S}{n}\right)^{0.495} + \frac{S \ln S}{n^{0.999}} + \frac{S \ln S \ln n}{n\gamma} + \sqrt{\frac{S \ln n \ln^2 S}{n\gamma}}, \tag{11}
$$

$$
|\bar{H}_{\text{emp}} - \bar{H}| \lesssim \frac{S^2}{n} + \left(\frac{S}{n}\right)^{0.495} + \frac{S \ln S}{n^{0.999}} + \frac{S \ln S \ln n}{n\gamma} + \sqrt{\frac{S \ln n \ln^2 S}{n\gamma}}. \tag{12}
$$

### 4.2 Proof of Theorem 2

By the concavity of entropy, the empirical entropy rate $\bar{H}_{\text{emp}}$ underestimates the truth $\bar{H}$ in expectation. On the other hand, the average codelength $\bar{L}$ of any lossless source code is at least $\bar{H}$ by Shannon's source coding theorem. As a result, $\bar{H} - \mathbb{E}[\bar{H}_{\text{emp}}] \leq \bar{L} - \mathbb{E}[\bar{H}_{\text{emp}}]$, and we may find a good lossless code to make the RHS small.

Since the conditional empirical distributions maximizes the likelihood for Markov chains (Lemma 13), we have

$$\mathbb{E}_P \left[ \frac{1}{n} \ln \frac{1}{Q_{X_1^n|X_0}(X_1^n|X_0)} \right] \geq \mathbb{E}_P \left[ \frac{1}{n} \ln \frac{1}{P_{X_1^n|X_0}(X_1^n|X_0)} \right] = \bar{H} \tag{13}$$

$$\geq \mathbb{E}_P \left[ \min_{P \in \mathcal{M}_2(S)} \frac{1}{n} \ln \frac{1}{P_{X_1^n|X_0}(X_1^n|X_0)} \right] = \mathbb{E}[\bar{H}_{\mathsf{emp}}] \tag{14}$$

where $\mathcal{M}_2(S)$ denotes the space of all first-order Markov chains with state $[S]$. Hence,

$$|\bar{H} - \mathbb{E}[\bar{H}_{\mathsf{emp}}]| \leq \inf_Q \sup_{P \in \mathcal{M}_2(S), x_0^n} \frac{1}{n} \ln \frac{P(x_1^n|x_0)}{Q(x_1^n|x_0)}. \tag{15}$$

The following lemma provides a non-asymptotic upper bound on the RHS of (15) and completes the proof of Theorem 2.

**Lemma 1.** *[3] Let $\mathcal{M}_2(S)$ denote the space of Markov chains with alphabet size $S$ for each symbol. Then, the worst case minimax redundancy is bounded as*

$$\inf_Q \sup_{P \in \mathcal{M}_2(S), x_0^n} \frac{1}{n} \ln \frac{P(x_1^n|x_0)}{Q(x_1^n|x_0)} \leq \frac{2S^2}{n} \ln \left( \frac{n}{S^2} + 1 \right) + \frac{(S^2+2) \ln 2}{n}. \tag{16}$$

### 4.3 Proof of Theorem 4

To prove the lower bound for Markov chains, we first introduce an auxiliary model, namely, the *independent Poisson* model and show that the sample complexity of the Markov chain model is lower bounded by that of the independent Poisson model. Then we apply the so-called method of fuzzy hypotheses [4, Theorem 2.15] (see also [1, Lemma 11]) to prove a lower bound for the independent Poisson model. We introduce the independent Poisson model below, which is parametrized by an $S \times S$ symmetric matrix $R$, an integer $n$ and a parameter $\lambda > 0$.

**Definition 1** (Independent Poisson model). *Given an $S \times S$ symmetric matrix $R = (R_{ij})$ with $R_{ij} \geq 0$ and a parameter $\lambda > 0$, under the independent Poisson model, we observe $X_0 \sim \pi = \pi(R)$, and an $S \times S$ matrix $C = (C_{ij})$ with independent entries distributed as $C_{ij} \sim \mathsf{Poi}(\lambda R_{ij})$, where*

$$\pi_i = \pi_i(R) = \frac{r_i}{r}, \quad r_i = \sum_{j=1}^S R_{ij}, \quad r = \sum_{i=1}^S r_i. \tag{17}$$

For each symmetric matrix $R$, by normalizing the rows we can define a transition matrix $T = T(R)$ of a *reversible* Markov chain with stationary distribution $\pi = \pi(R)$. Upon observing the Poisson matrix $C$, the functional to be estimated is the entropy rate $\bar{H}$ of $T(R)$. Given $\tau > 0$ and $\gamma, q \in (0,1)$, define the following collection of symmetric matrices:

$$\mathcal{R}(S, \gamma, \tau, q) = \left\{ R \in \mathbb{R}_+^{S \times S} : R = R^\top, \gamma^*(T) \geq \gamma, \sum_{i,j} R_{ij} \geq \tau, \pi_{\min} \geq q \right\}, \tag{18}$$

where $\pi_{\min} = \min_i \pi_i$. The reduction to independent Poisson model is summarized below:

**Lemma 2.** *If there exists an estimator $\hat{H}_1$ for the Markov chain model with parameter $n$ such that $\mathbb{P}(|\hat{H}_1 - \bar{H}| \geq \epsilon) \leq \delta$ under any $T \in \mathcal{M}_{2,\mathrm{rev}}(S, \gamma)$, then there exists another estimator $\hat{H}_2$ for the independent Poisson model with parameter $\lambda = \frac{4n}{\tau}$ such that*

$$\sup_{R \in \mathcal{R}(S, \gamma, \tau, q)} \mathbb{P} \left( |\hat{H}_2 - \bar{H}(T(R))| \geq \epsilon \right) \leq \delta + 2Sn^{-\frac{c_3^2}{4+10c_3}} + Sn^{-c_3/2}, \tag{19}$$

*provided $q \geq \frac{c_3 \ln n}{n\gamma}$, where $c_3 \geq 20$ is a universal constant.*

To prove the lower bound for the independent Poisson model, the goal is to construct two symmetric random matrices (whose distributions serve as the priors), such that (a) they are sufficiently concentrated near the desired parameter space $\mathcal{R}(S, \gamma, \tau, q)$ for properly chosen parameters

$\gamma, \tau, q$; (b) their entropy rates are separated; (c) the induced marginal laws of the sufficient statistic $\mathbf{C} = X_0 \cup \{C_{ij} + C_{ji} : i \neq j, 1 \leq i \leq j \leq S\} \cup \{C_{ii} : 1 \leq i \leq S\}$ are statistically indistinguishable. The prior construction in Definition 4 satisfies all these three properties (cf. Lemmas 10, 11, 12), and thereby lead to the following theorem:

**Theorem 6.** *If $n \geq \frac{S^2}{\ln S}, \ln n \ll \frac{S}{(\ln S)^2}, \gamma^* \leq 1 - C_2 \sqrt{\frac{S \ln^3 S}{n}}$, we have*

$$\liminf_{S \to \infty} \inf_{\hat{H}} \sup_{R \in \mathcal{R}(S, \gamma^*, \tau, q)} \mathbb{P}\left(|\hat{H} - \bar{H}| \geq C_1 \frac{S^2}{n \ln S}\right) \geq \frac{1}{2} \tag{20}$$

*where $\tau = S, q = \frac{1}{5\sqrt{n \ln S}}$, and $C_1, C_2 > 0$ are two universal constants.*

## 5  Application: Fundamental limits of language modeling

In this section, we apply entropy rate estimators to estimate the fundamental limits of language modeling. A language model specifies the joint probability distribution of a sequence of words, $Q_{X^n}(x^n)$. It is common to use a $(k-1)$th-order Markov assumption to train these models, using sequences of $k$ words (also known as $k$-grams,[3] sometimes with Latin prefixes *unigrams*, *bigrams*, *etc.*), with values of $k$ of up to 5 [2]. A commonly used metric to measure the efficacy of a model $Q_{X^n}$ is the *perplexity* (whose logarithm is called the *cross-entropy rate*):

$$\text{perplexity}_Q\left(X^n\right) = \sqrt[n]{\frac{1}{Q_{X^n}(X^n)}}.$$

If a language is modeled as a stationary and ergodic stochastic process with entropy rate $\bar{H}$, and $X^n$ is drawn from the language with true distribution $P_{X^n}$, then [2]

$$\bar{H} \leq \liminf_{n \to \infty} \frac{1}{n} \log \frac{1}{Q_{X^n}(X^n)} = \liminf_{n \to \infty} \log\left[\text{perplexity}_Q\left(X^n\right)\right],$$

with equality when $Q = P$. In this section, all logarithms are with respect to base 2 and all entropy are measured in bits.

The entropy rate of the English language is of significant interest to language model researchers: since $2^{\bar{H}}$ is a tight lower bound on perplexity, this quantity indicates how close a given language model is to the optimum. Several researchers have presented estimates in bits per character [3, 9, 5]; because language models are trained on words, these estimates are not directly relevant to the present task. In one of the earliest papers on this topic, Claude Shannon [3] gave an estimate of 11.82 bits per word. This latter figure has been comprehensively beaten by recent models; for example, [2] achieved a perplexity corresponding to a cross-entropy rate of 4.55 bits per word.

To produce an estimate of the entropy rate of English, we used two well-known linguistic corpora: the Penn Treebank (PTB) and Google's One Billion Words (1BW) benchmark. Results based on these corpora are particularly relevant because of their widespread use in training models. We used the conditional approach proposed in this paper with the JVHW estimator describe in Section D. The PTB corpus contains about $n \approx 1.2$ million words, of which $S \approx 47,000$ are unique. The 1BW corpus contains about $n \approx 740$ million words, of which $S \approx 2.4$ million are unique.

We estimate the conditional entropy $H(X_k|X^{k-1})$ for $k = 1, 2, 3, 4$, and our results are shown in Figure 1. The estimated conditional entropy $\hat{H}(X_k|X^{k-1})$ provides us with a refined analysis of the intrinsic uncertainty in language prediction with context length of only $k-1$. For 4-grams, using the JVHW estimator on the 1BW corpus, our estimate is 3.46 bits per word. With current state-of-the-art models trained on the 1BW corpus having an cross-entropy rate of about 4.55 bits per word [2], this indicates that language models are still at least 0.89 bits per word away from the fundamental limit. (Note that since $H(X_k|X^{k-1})$ is decreasing in $k$, $H(X_4|X^3) > \bar{H}$.) Similarly, for the much smaller PTB corpus, we estimate an entropy rate of 1.50 bits per word, compared to state-of-the-art models that achieve a cross-entropy rate of about 5.96 bits per word [4], at least 4.4 bits away from the fundamental limit.

More detailed analysis, e.g., the accuracy of the JVHW estimates, is shown in the Appendix E.

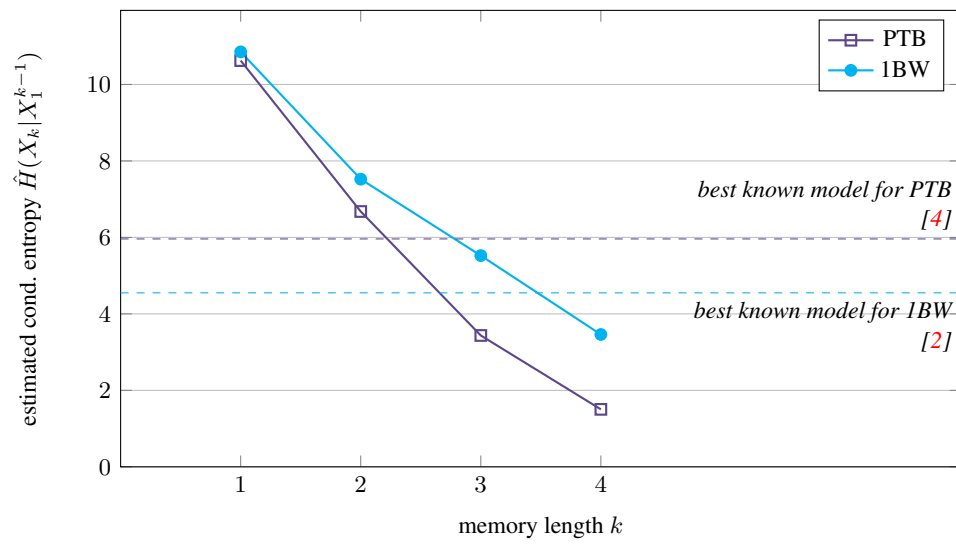

Figure 1: Estimates of conditional entropy based on linguistic corpora

## Footnotes

[1]This exceeds the estimated vocabulary of the English language partly because different forms of a word count as different words in language models, and partly because of edge cases in tokenization, the automatic splitting of text into "words".

[2]The asymptotic results in this section are interpreted by parameterizing $n = n_S$ and $\gamma = \gamma_S$ and $S \to \infty$ subject to the conditions of each theorem.

[3] In the language modeling literature these are typically known as $n$-grams, but we use $k$ to avoid conflict with the sample size.

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
