[Supplementary Material]

# A  Proof of Theorem 1

## A.1  Concentration of $\hat{H}_{\text{opt}}$ and $\hat{H}_{\text{emp}}$

The performance of $\hat{H}_{\text{opt}}$ and $\hat{H}_{\text{emp}}$ in terms of Shannon entropy estimation is collected in the following lemma.

**Lemma 3.** *Suppose $\alpha = 0.001, \alpha' = 0.01$ and one observes $n$ i.i.d. samples $X_1, X_2, \ldots, X_n \overset{\text{i.i.d.}}{\sim} P$. Then, there exists an entropy estimator $\hat{H}_{\text{opt}} = \hat{H}_{\text{opt}}(X_1, \ldots, X_n) \in [0, \ln S]$ such that for any $t > 0$,*

$$P\left(|\hat{H}_{\text{opt}} - H(P)| \geq t + c_2 \frac{S}{n \ln S}\right) \leq 2\exp\left(-c_1 t^2 n^{1-\alpha}\right), \tag{21}$$

*where $c_1, c_2 > 0$ are universal constants, and $H(P)$ is the Shannon entropy. Moreover, the empirical entropy $\hat{H}_{\text{emp}} = \hat{H}_{\text{emp}}(X_1, X_2, \ldots, X_n) \in [0, \ln S]$ satisfies, for any $t > 0$,*

$$P\left(|\hat{H}_{\text{emp}} - H(P)| \geq t + c_2 \frac{S}{n}\right) \leq 2\exp\left(-c_1 t^2 n^{1-\alpha}\right). \tag{22}$$

*Consequently, for any $\beta > 0$,*

$$P\left(|\hat{H}_{\text{opt}} - H(P)| \geq \frac{c_2 S}{n \ln S} + \sqrt{\frac{\beta}{c_1 n^{1-\alpha'}}}\right) \leq \frac{2}{n^\beta}, \tag{23}$$

*and*

$$P\left(|\bar{H}_{\text{emp}}) - H(P)| \geq \frac{c_2 S}{n} + \sqrt{\frac{\beta}{c_1 n^{1-\alpha'}}}\right) \leq \frac{2}{n^\beta}. \tag{24}$$

*Proof.* The part pertaining to the concentration of $\hat{H}_{\text{opt}}$ follows from [45, 19, 1]. The part pertaining to the empirical entropy follows from [2],[31, Proposition 1],[20, Eqn. (88)]. □

## A.2  Analysis of $\bar{H}_{\text{opt}}$ and $\bar{H}_{\text{emp}}$

Next we define two events that ensure the proposed entropy rate estimator $\bar{H}_{\text{opt}}$ and the empirical entropy rate $\bar{H}_{\text{emp}}$ is accurate, respectively:

**Definition 2** ("Good" event in estimation). *Let $0 < c_4 < 1$ and $c_3 \geq 20$ be some universal constants. We take $c_4 = 0.001$.*

*1. For every $i, 1 \leq i \leq S$, define the event*

$$\mathcal{E}_i = \left\{|\hat{\pi}_i - \pi_i| \leq c_3 \max\left\{\frac{\ln n}{n\gamma}, \sqrt{\frac{\pi_i \ln n}{n\gamma}}\right\}\right\} \tag{25}$$

*2. For every $i \in [S]$ such that $\pi_i \geq n^{c_4-1} \vee 100 c_3^2 \frac{\ln n}{n\gamma}$, define the event $\mathcal{H}_i$ as*

$$|\hat{H}_{\text{opt}}(W_{i1}, W_{i2}, \ldots, W_{im}) - H_i| \leq \frac{c_2 S}{m \ln S} + \sqrt{\frac{\beta}{c_1 m^{1-\alpha'}}}, \tag{26}$$

*for all $m$ such that $n\pi_i - c_3\sqrt{\frac{n\pi_i \ln n}{\gamma}} \leq m \leq n\pi_i + c_3\sqrt{\frac{n\pi_i \ln n}{\gamma}}$, where $\beta = \frac{c_3^2}{4+10c_3}$, $c_1, c_2, \alpha'$ are from Lemma 3.*

*Finally, define the "good" event as the intersection of all the events above:*

$$\mathcal{G}_{\text{opt}} \triangleq \left(\bigcap_{i \in [S]} \mathcal{E}_i\right) \cap \left(\bigcap_{i: \pi_i \geq n^{c_4-1} \vee 100 c_3^2 \frac{\ln n}{n\gamma}} \mathcal{H}_i\right). \tag{27}$$

*Analogously, we define the "good" event $\mathcal{G}_{\text{emp}}$ for the empirical entropy rate $\bar{H}_{\text{emp}}$ in a similar fashion with (26) replaced by*

$$|\hat{H}_{\text{emp}}(W_{i1}, W_{i2}, \ldots, W_{im}) - H_i| \leq \frac{c_2 S}{m} + \sqrt{\frac{\beta}{c_1 m^{1-\alpha'}}}. \tag{28}$$

The following lemma shows that the "good" events defined in Definition 2 indeed occur with high probability.

**Lemma 4.** *Both $\mathcal{G}_{\text{opt}}$ and $\mathcal{G}_{\text{emp}}$ in Definition 2 occur with probability at least*

$$1 - \frac{2S}{n^\beta} - \frac{4c_3(10)^\beta}{9^\beta}\frac{S}{n^{c_4(\beta-1)}}, \tag{29}$$

*where $\beta = \frac{c_3^2}{4+10c_3}, c_3 \geq 20$.*

*Proof of Theorem 5.* Pick $c_4 = 0.001, \alpha' = 0.01$. We write

$$\bar{H} = \sum_{i=1}^{S} \pi_i H_i, \tag{30}$$

$$\bar{H}_{\text{opt}} = \sum_{i=1}^{S} \hat{\pi}_i \hat{H}_i, \tag{31}$$

where $H_i = H(X_2|X_1 = i), \hat{H}_i = \hat{H}_{\text{opt}}(\mathbf{X}^{(i)}) = \hat{H}_{\text{opt}}(W_{i1}, \ldots, W_{in_i})$. Write

$$\bar{H}_{\text{opt}} - \bar{H} = \underbrace{\sum_{i=1}^{S} \pi_i \left(\hat{H}_i - H_i\right)}_{E_1} + \underbrace{\sum_{i=1}^{S} \hat{H}_i(\hat{\pi}_i - \pi_i)}_{E_2}. \tag{32}$$

Next we bound the two terms separately under the condition that the "good" event $\mathcal{G}_{\text{opt}}$ in Definition 2 occurs.

Note that the function $\pi_i \mapsto n\pi_i - c_3\sqrt{\frac{n\pi_i \ln n}{\gamma}}$ is an increasing function when $\pi_i \geq \frac{100c_3^2 \ln n}{n\gamma}$. Thus we have

$$n\pi_i - c_3\sqrt{\frac{n\pi_i \ln n}{\gamma}} = n\pi_i\left(1 - c_3\sqrt{\frac{\ln n}{n\pi_i\gamma}}\right) \geq \frac{9}{10}n\pi_i, \tag{33}$$

whenever $\pi_i \geq \frac{100c_3^2 \ln n}{n\gamma}$.

Let $\epsilon(m) \triangleq \frac{c_2 S}{m \ln S} + \sqrt{\frac{\beta}{c_1 m^{1-\alpha'}}}$, which is decreasing in $m$. Let $n_i^{\pm} \triangleq n\pi_i \pm c_3 \max\left\{\frac{\ln n}{\gamma}, \sqrt{\frac{n\pi_i \ln n}{\gamma}}\right\}$. Note that for each $i \in [S]$,

$$\left\{|\hat{H}_i - H_i| \leq \epsilon(n_i)\right\} \supset \left\{|\hat{H}_i - H_i| \leq \epsilon(n_i), |\hat{\pi}_i - \pi_i| \leq \max\left\{\frac{\ln n}{n\gamma}, \sqrt{\frac{\pi_i \ln n}{n\gamma}}\right\}\right\}$$

$$= \left\{|\hat{H}_{\text{opt}}(W_{i1}, \ldots, W_{in_i}) - H_i| \leq \epsilon(n_i), n_i^- \leq n_i \leq n_i^+\right\}$$

$$\supset \bigcap_{m=n_i^-}^{n_i^+} \{|\hat{H}_{\text{opt}}(W_{i1}, \ldots, W_{im}) - H_i| \leq \epsilon(m)\}.$$

The key observation is that for each fixed $m$, $W_{i1}, \ldots, W_{im}$ are i.i.d. as $T_i$.[4] Taking the intersection over $i \in [S]$, we have

$$\left\{|\hat{H}_i - H_i| \leq \epsilon(n_i), \ i = 1, \ldots, S\right\} \supset \mathcal{G}_{\text{opt}}.$$

Therefore, on the event $\mathcal{G}_{\text{opt}}$, we have

$$
\begin{aligned}
|E_1| &\le \sum_{i=1}^{S} \pi_i |\hat{H}_i - H_i| \\
&\le \sum_{i:\pi_i \ge n^{c_4-1} \vee 100 c_3^2 \frac{\ln n}{n\gamma}} \pi_i |\hat{H}_i - H_i| + \sum_{i:\pi_i \le n^{c_4-1} \vee 100 c_3^2 \frac{\ln n}{n\gamma}} \pi_i |\hat{H}_i - H_i| \\
&\overset{(33)}{\le} \sum_{i:\pi_i \ge n^{c_4-1} \vee 100 c_3^2 \frac{\ln n}{n\gamma}} \pi_i \left( \frac{c_2 S}{0.9 n \pi_i \ln S} + \sqrt{\frac{\beta}{c_1 (0.9 n \pi_i)^{1-\alpha'}}} \right) \\
&\qquad + \sum_{i:\pi_i \le n^{c_4-1} \vee 100 c_3^2 \frac{\ln n}{n\gamma}} \pi_i \ln S \\
&\lesssim \frac{S^2}{n \ln S} + \left( \frac{S}{n} \right)^{\frac{1-\alpha'}{2}} + \frac{S \ln S}{n^{1-c_4}} \vee \frac{S \ln S \ln n}{n\gamma},
\end{aligned}
\tag{34}
$$

where the last step follows from (33) and the fact that $\sum_{i\in[S]} \pi_i^\alpha \le S^{1-\alpha}$ for any $\alpha \in [0,1]$. As for $E_2$, on the event $\mathcal{G}_{\text{opt}}$, we have

$$
|E_2| \le \sum_{i=1}^{S} \hat{H}_i |\hat{\pi}_i - \pi_i| \le \ln S \sum_{i=1}^{S} c_3 \max\left\{ \frac{\ln n}{n\gamma}, \sqrt{\frac{\pi_i \ln n}{n\gamma}} \right\} \lesssim \frac{S \ln S \ln n}{n\gamma} \vee \sqrt{\frac{S \ln n \ln^2 S}{n\gamma}}.
\tag{35}
$$

Combining (34) and (35), and using Lemma 4, completes the proof of (11). The proof of (12) follows entirely analogously with $\mathcal{G}_{\text{opt}}$ replaced by $\mathcal{G}_{\text{emp}}$. □

## B   Proof of Theorem 3

We first prove Theorem 3, which quantifies the performance limit of the empirical entropy rate. Lemma 13 in Section F shows that

$$
\bar{H}_{\text{emp}} = \min_{P \in \mathcal{M}_2(S)} \frac{1}{n} \ln \frac{1}{P_{X_1^n|X_0}(X_1^n|X_0)},
\tag{36}
$$

where $\mathcal{M}_2(S)$ denotes the set of all Markov chain transition matrices with state space $\mathcal{X}$ of size $S$. Since

$$
\bar{H} = \mathbb{E}_P \left[ \frac{1}{n} \ln \frac{1}{P_{X_1^n|X_0}(X_1^n|X_0)} \right],
\tag{37}
$$

we know $\bar{H} - \mathbb{E}[\bar{H}_{\text{emp}}] \ge 0$.

We specify the true distribution $P_{X_0^n}(x_0^n)$ to be the i.i.d. product distribution $\prod_{i=0}^{n} P(x_i)$, and it suffices to lower bound

$$
\mathbb{E}_P \left[ \frac{1}{n} \ln \frac{1}{P_{X_1^n|X_0}(X_1^n|X_0)} - \min_{P \in \mathcal{M}_2(S)} \frac{1}{n} \ln \frac{1}{P_{X_1^n|X_0}(X_1^n|X_0)} \right]
\tag{38}
$$

$$
= H(P) - \mathbb{E}_P \left[ H(\hat{P}_{X_1 X_2}) - H(\hat{P}_{X_1}) \right]
\tag{39}
$$

$$
= \left( H(P_{X_1 X_2}) - \mathbb{E}_P[H(\hat{P}_{X_1 X_2})] \right) - (H(P_{X_1}) - \mathbb{E}_P[H(\hat{P}_{X_1})]),
\tag{40}
$$

where $\hat{P}_{X_1 X_2}$ is the empirical distribution of the counts $\{(x_i, x_{i+1}) : 0 \le i \le n-1\}$, and $\hat{P}_{X_1}$ is the marginal distribution of $\hat{P}_{X_1 X_2}$.

It was shown in [20] that for any $P_{X_1}$,

$$
0 \le H(P_{X_1}) - \mathbb{E}_P[H(\hat{P}_{X_1})] \le \ln\left( 1 + \frac{S-1}{n} \right).
\tag{41}
$$

Now, choosing $P_{X_1}$ to be the uniform distribution, we have

$$\mathbb{E}_P\left[\frac{1}{n}\ln\frac{1}{P_{X_1^n|X_0}(X_1^n|X_0)} - \min_{P\in\mathcal{M}_2(S)}\frac{1}{n}\ln\frac{1}{P_{X_1^n|X_0}(X_1^n|X_0)}\right] \tag{42}$$

$$\geq \ln(S^2) - \ln n - \ln\left(1 + \frac{S-1}{n}\right) \tag{43}$$

$$\geq \ln\left(\frac{S^2}{n+S-1}\right), \tag{44}$$

where we have used the fact that the uniform distribution on $S$ elements has entropy $\ln(S)$, and it maximizes the entropy among all distribution supported on $S$ elements.

## C   Proof of Theorem 4

We first show that Lemma 2 and Theorem 6 imply Theorem 4. Firstly, Theorem 6 shows that as $S \to \infty$, under Poisson independent model,

$$\inf_{\hat{H}} \sup_{R\in\mathcal{R}(S,\gamma^*,\tau,q)} \mathbb{P}\left(|\hat{H} - \bar{H}| \geq C_1\frac{S^2}{n\ln S}\right) \geq \frac{1}{2} - o(1) \tag{45}$$

where $\tau = S, q = \frac{1}{5\sqrt{n\ln S}}$. Moreover, since a larger $\gamma^*$ results in a smaller set of parameters for all models, we may always assume that $\gamma^* = 1 - C_2\sqrt{\frac{S\ln^3 S}{n}}$. For this choice of $\gamma^*$, the assumption $n \geq \frac{S^2}{\ln S}$ ensures $q = \frac{1}{5\sqrt{n\ln S}} \geq \frac{c_3\ln n}{n\gamma^*}$, and thus Lemma 2 implies

$$\inf_{\hat{H}} \sup_{T\in\mathcal{M}_{2,\text{rev}}(S,\gamma^*)} \mathbb{P}\left(|\hat{H} - \bar{H}| \geq C_1\frac{S^2}{n\ln S}\right) \geq \frac{1}{2} - o(1) - 2Sn^{-\frac{c_3^2}{4+10c_3}} - Sn^{-c_3/2} = \frac{1}{2} - o(1)$$

under the Markov chain model, completing the proof of Theorem 4.

### C.1   Proof of Lemma 2

We introduce an additional auxiliary model, namely, the *independent multinomial* model, and show that the sample complexity of the Markov chain model is lower bounded by that of the independent multinomial model (Lemma 5), which is further lower bounded by that of the independent Poisson model (Lemma 6). To be precise, we use the notation $P_{\text{MC}}, P_{\text{IM}}, P_{\text{IP}}$ to denote the probability measure corresponding to the three models respectively.

#### C.1.1   Reduction from Markov chain to independent multinomial

**Definition 3** (Independent multinomial model). *Given a stationary reversible Markov chain with transition matrix $T = (T_{ij}) \in \mathcal{M}_{2,\text{rev}}(S)$, stationary distribution $\pi_i, i \in [S]$ and absolute spectral gap $\gamma^*$. Fix an integer $n \geq 0$. Under the independent multinomial model, the statistician observes $X_0 \sim \pi$, and the following arrays of independent random variables*

$$\begin{array}{cccc}
W_{11}, & W_{12}, & \dots, & W_{1m_1} \\
W_{21}, & W_{22}, & \dots, & W_{2m_2} \\
\vdots, & \vdots, & \ddots, & \vdots \\
W_{S1}, & W_{S2}, & \dots, & W_{Sm_S}
\end{array}$$

*where the number of observations in the ith row is $m_i = \lceil n\pi_i + c_3\max\left\{\frac{\ln n}{\gamma^*}, \sqrt{\frac{n\pi_i\ln n}{\gamma^*}}\right\}\rceil$ for some constant $c_3 \geq 20$, and within the ith row the random variables $W_{i1}, W_{i2}, \dots, W_{im_i} \overset{i.i.d.}{\sim} T_i$.*

Equivalently, the observations can be summarized into the following (sufficient statistic) $S \times S$ matrix $C = (C_{ij})$, where each row is independently distributed $\text{multi}(m_i, T_i)$, hence the name of independent multinomial model.

The following lemma relates the independent multinomial model to the Markov chain model:

**Lemma 5.** *If there exists an estimator $\hat{H}_1$ under the Markov chain model with parameter $n$ such that*

$$\sup_{T\in\mathcal{M}_{2,rev}(S,\gamma^*)} P_{\mathsf{MC}}\left(|\hat{H}_1 - \bar{H}| \geq \epsilon\right) \leq \delta, \tag{46}$$

*then there exists another estimator $\hat{H}_2$ under the independent multinomial model with parameter $n$ such that*

$$\sup_{T\in\mathcal{M}_{2,rev}(S,\gamma^*)} P_{\mathsf{IM}}\left(|\hat{H}_2 - \bar{H}| \geq \epsilon\right) \leq \delta + \frac{2S}{n^\beta}, \tag{47}$$

*where $\beta = \frac{c_3^2}{4+10c_3} \geq 1$, and $c_3$ is the constant in Definition 3.*

### C.1.2 Reduction from independent multinomial to independent Poisson

For the reduction from the independent multinomial model to the independent Poisson model, we have the following lemma. Note that

$$\bar{H}(T(R)) = \sum_{1\leq i,j\leq S} \pi_i \frac{R_{ij}}{\sum_{j=1}^S R_{ij}} \ln \frac{\sum_{j=1}^S R_{ij}}{R_{ij}} \tag{48}$$

$$= \frac{1}{r} \sum_{1\leq i,j\leq S} R_{ij} \ln \frac{r_i}{R_{ij}} \tag{49}$$

$$= \frac{1}{r} \left( \sum_{1\leq i,j\leq S} R_{ij} \ln \frac{1}{R_{ij}} + \sum_{i=1}^S r_i \ln r_i \right). \tag{50}$$

**Lemma 6.** *If there exists an estimator $\hat{H}_1$ for the independent multinomial model with parameter $n$ such that*

$$\sup_{T\in\mathcal{M}_{2,rev}(S,\gamma)} P_{\mathsf{IM}}\left(|\hat{H}_1 - \bar{H}| \geq \epsilon\right) \leq \delta, \tag{51}$$

*then there exists another estimator $\hat{H}_2$ for the independent Poisson model with parameter $\lambda = \frac{4n}{\tau}$ such that*

$$\sup_{R\in\mathcal{R}(S,\gamma,\tau,q)} P_{\mathsf{IP}}\left(|\hat{H}_2 - \bar{H}(T(R))| \geq \epsilon\right) \leq \delta + Sn^{-c_3/2}, \tag{52}$$

*provided $q \geq \frac{c_3 \ln n}{n\gamma}$, where $c_3 \geq 20$ is the constant in Definition 3.*

### C.2 Proof of Theorem 6

Now our task is reduced to lower bounding the sample complexity of the independent Poisson model. The general strategy is the so-called method of fuzzy hypotheses, which is an extension of LeCam's two-point methods. The following version is adapted from [40, Theorem 2.15] (see also [14, Lemma 11]).

**Lemma 7.** *Let $\mathbf{Z}$ be a random variable distributed according to $P_\theta$ for some $\theta \in \Theta$. Let $\mu_1, \mu_2$ be a pair of probability measures (not necessarily supported on $\Theta$). Let $\hat{f} = \hat{f}(\mathbf{Z})$ be an arbitrary estimator of the functional $f(\theta)$ based on the observation $\mathbf{Z}$. Suppose there exist $\zeta \in \mathbb{R}, \Delta > 0, 0 \leq \beta_1, \beta_2 < 1$ such that*

$$\mu_1(\theta \in \Theta : f(\theta) \leq \zeta - \Delta) \geq 1 - \beta_1 \tag{53}$$
$$\mu_2(\theta \in \Theta : f(\theta) \geq \zeta + \Delta) \geq 1 - \beta_2. \tag{54}$$

*Then*

$$\inf_{\hat{f}} \sup_{\theta\in\Theta} \mathbb{P}_\theta\left(|\hat{f} - f(\theta)| \geq \Delta\right) \geq \frac{1 - \mathsf{TV}(F_1, F_2) - \beta_1 - \beta_2}{2}, \tag{55}$$

*where $F_i = \int P_\theta \mu_i(d\theta)$ is the marginal distributions of $\mathbf{Z}$ induced by the prior $\mu_i$, for $i = 1, 2$, and $\mathsf{TV}(F_1, F_2) = \frac{1}{2}\int |dF_1 - dF_2|$ is the total variation distance between distributions $F_1$ and $F_2$.*

To apply this method for the independent Poisson model, the parameter is the $S \times S$ symmetric matrix $R$, the function to be estimated is $\bar{H} = \bar{H}(T(R))$, the observation (sufficient statistic for $R$) is
$$\mathbf{C} = X_0 \cup \{C_{ij} + C_{ji} : i \neq j, 1 \leq i \leq j \leq S\} \cup \{C_{ii} : 1 \leq i \leq S\}.$$
The goal is to construct two symmetric random matrices (whose distributions serve as the priors), such that

(a) they are sufficiently concentrated near the desired parameter space $\mathcal{R}(S, \gamma, \tau, q)$ for properly chosen parameters $\gamma, \tau, q$;

(b) the entropy rates have different values;

(c) the induced marginal laws of $\mathbf{C}$ are statistically inseparable.

To this end, we need the following results (cf. [45, Proof of Proposition 3]):

**Lemma 8.** *Let*
$$\phi(x) \triangleq x \ln \frac{1}{x}, \quad x \in [0, 1].$$
*Let $c > 0, D > 100$ and $0 < \eta_0 < 1$ be some absolute constants. For any $\alpha \in (0, 1), \eta \in (0, \eta_0)$, there exist random variables $U, U'$ supported on $[0, \alpha\eta^{-1}]$ such that*

$$\mathbb{E}[\phi(U)] - \mathbb{E}[\phi(U')] \geq c\alpha \tag{56}$$

$$\mathbb{E}[U^j] = \mathbb{E}[U'^j], \quad j = 1, 2, \ldots, \left\lceil \frac{D}{\sqrt{\eta}} \right\rceil \tag{57}$$

$$\mathbb{E}[U] = \mathbb{E}[U'] = \alpha. \tag{58}$$

**Lemma 9** ([45, Lemma 3]). *Let $V_1$ and $V_2$ be random variables taking values in $[0, M]$. If $\mathbb{E}[V_1^j] = \mathbb{E}[V_2^j]$, $j = 1, \ldots, L$, then*

$$\mathsf{TV}(\mathbb{E}[\mathsf{Poi}(V_1)], \mathbb{E}[\mathsf{Poi}(V_2)]) \leq \left( \frac{2eM}{L} \right)^L. \tag{59}$$

*where $\mathbb{E}[\mathsf{Poi}(V)] = \int \mathsf{Poi}(\lambda) P_V(d\lambda)$ denotes the Poisson mixture with respect to the distribution of a positive random variable $V$.*

Now we are ready to define the priors $\mu_1, \mu_2$ for the independent Poisson model. For simplicity, we assume the cardinality of the state space is $S + 1$ and introduce a new state 0:

**Definition 4** (Prior construction). *Suppose $n \geq \frac{S^2}{\ln S}$. Set*

$$\alpha = \frac{S}{n \ln S} \leq \frac{1}{S} \tag{60}$$

$$\frac{1}{\eta} = (d_1 \ln S)^2 \tag{61}$$

$$L = \left\lceil \frac{D}{\sqrt{\eta}} \right\rceil, \tag{62}$$

*where $d_1 = \frac{D}{8e^2}$, and $D > 0$ is the constant in Lemma 8.*

*Recall the random variables $U, U'$ are introduced in Lemma 8. We use a construction that is akin to that studied in [4]. Define $S \times S$ symmetric random matrices $\mathbf{U} = (U_{ij})$ and $\mathbf{U}' = (U'_{ij})$, where $\{U_{ij} : 1 \leq i \leq j \leq S\}$ be i.i.d. copies of $U$ and $\{U'_{ij} : 1 \leq i \leq j \leq S\}$ be i.i.d. copies of $U'$, respectively. Let*

$$\mathbf{R} = \left[ \begin{array}{c|c} b & a \cdots a \\ \hline a & \\ \vdots & \mathbf{U} \\ a & \end{array} \right], \qquad \mathbf{R}' = \left[ \begin{array}{c|c} b & a \cdots a \\ \hline a & \\ \vdots & \mathbf{U}' \\ a & \end{array} \right], \tag{63}$$

*where*

$$a = \sqrt{\alpha S}, \quad b = S. \tag{64}$$

*Let $\mu_1$ and $\mu_2$ be the laws of $\mathbf{R}$ and $\mathbf{R}'$, respectively. The parameters $\gamma, \tau, q$ will be chosen later, and we set $\lambda = \frac{4n}{\tau}$ in the independent Poisson model (as in Lemma 6).*

The construction of this pair of priors achieves the following three goals:

**(a) Statistical indistinguishablility.** Note that the distributions of the first row and column of $\mathbf{R}$ and $\mathbf{R}'$ are identical. Hence the sufficient statistics are $X_0$ and $\mathbf{C} = \{C_{ij} + C_{ji} : i \neq j, 1 \leq i \leq j \leq S\} \cup \{C_{ii}, 1 \leq i \leq S\}$. Denote its the marginal distribution as $F_i$ under the prior $\mu_i$, for $i = 1, 2$. The following lemma shows that the distributions of the sufficient statistic are indistinguishable:

**Lemma 10.** *For $n \geq \frac{S^2}{\ln S}$, we have* $\mathsf{TV}(F_1, F_2) = o(1)$ *as $S \to \infty$.*

**(b) Functional value separation.** Under the two priors $\mu_1, \mu_2$, the corresponding entropy rates of the independent Poisson model differ by a constant factor of $\frac{S^2}{n \ln S}$. Here we explain the intuition: in view of (50), for $\phi(x) = -x \ln x$ we have

$$\bar{H}(T(\mathbf{R})) = \frac{1}{r} \left( \sum_{i,j=0}^{S} \phi(R_{ij}) - \sum_{i=0}^{S} \phi(r_i) \right) \tag{65}$$

where $r_i = \sum_{j=0}^{S} R_{ij}$ and $r = \sum_{i,j=0}^{S} R_{ij}$; similarly,

$$\bar{H}(T(\mathbf{R}')) = \frac{1}{r'} \left( \sum_{i,j=0}^{S} \phi(R'_{ij}) - \sum_{i=0}^{S} \phi(r'_i) \right).$$

We will show that both $r$ and $r'$ are close to their common mean $b + 2aS + S^2\alpha = S(1 + \sqrt{\alpha S})^2 \approx S$. Furthermore, $r_i$ and $r'_i$ also concentrate on their common mean. Thus, in view of Lemma 8, we have

$$|\bar{H}(T(\mathbf{R})) - \bar{H}(T(\mathbf{R}'))| \approx S|\mathbb{E}[\phi(U)] - \mathbb{E}[\phi(U')]| = \Omega(S\alpha) = \Omega\left(\frac{S^2}{n \ln S}\right). \tag{66}$$

The precise statement is summarized in the following lemma:

**Lemma 11.** *Assume that $n \geq \frac{S^2}{\ln S}$ and $\ln n \ll \frac{S}{\ln^2 S}$. There exist universal constants $C_1 > 0$ and some $\zeta \in \mathbb{R}$, such that as $S \to \infty$,*

$$\mathbb{P}\left( \bar{H}(T(\mathbf{R})) \geq \zeta + C_1 \frac{S^2}{n \ln S} \right) = 1 - o(1),$$

$$\mathbb{P}\left( \bar{H}(T(\mathbf{R}')) \leq \zeta - C_1 \frac{S^2}{n \ln S} \right) = 1 - o(1).$$

**(c) Concentration on parameter space.** Although the random matrices $\mathbf{R}$ and $\mathbf{R}'$ may take values outside the desired space $\mathcal{R}(S, \gamma, \tau, q)$, we show that most of the mass is concentrated on this set with appropriately chosen parameters. The following lemma, which is the core argument of the lower bound, makes this statement precise.

**Lemma 12.** *Assume that $n \geq \frac{S^2}{\ln S}$. There exist universal constants $C > 0$, such that as $S \to \infty$,*

$$\mathbb{P}\left( \mathbf{R} \in \mathcal{R}(S, \gamma, \tau, q) \right) = 1 - o(1),$$

$$\mathbb{P}\left( \mathbf{R}' \in \mathcal{R}(S, \gamma, \tau, q) \right) = 1 - o(1),$$

*where $\gamma = 1 - C_2\sqrt{\frac{S \ln^3 S}{n}}$, $\tau = S$, and $q = \frac{1}{5\sqrt{n \ln S}}$.*

Fitting Lemma 10, Lemma 11 and Lemma 12 into the main Lemma 7, the following minimax lower bound holds for the independent Poisson model.

*Proof of Theorem 6.* For the choice of $\zeta$ and $\Delta = C_1 \frac{S^2}{n \ln S}$ in Lemma 11, a combination of Lemma 11 and Lemma 12 gives

$$\mathbb{P}\left( \mathbf{R} \in \mathcal{R}(S, \gamma^*, \tau, q), \bar{H}(T(\mathbf{R})) \geq \zeta + C_1 \frac{S^2}{n \ln S} \right) = 1 - o(1) \tag{67}$$

as $S \to \infty$, so that $\beta_1 = o(1)$. Similarly, $\beta_2 = o(1)$. By Lemma 10, we have $\mathsf{TV}(F_1, F_2) = o(1)$. Now Theorem 6 follows from Lemma 7 directly. $\square$

# D Experiments

The entropy rate estimator we proposed in this paper that achieves the minimax rates can be viewed as a *conditional* approach; in other words, we apply a Shannon entropy estimator for observations corresponding to each state, and then average the estimates using the empirical frequency of the states. More generally, for any estimator $\hat{H}$ of the Shannon entropy from i.i.d. data, the conditional approach follows the idea of

$$\bar{H}_{\text{Cond}} = \sum_{i=1}^{S} \hat{\pi}_i \hat{H}(\mathbf{X}^{(i)}), \tag{68}$$

where $\hat{\pi}$ is the empirical marginal distribution. We list several choices of $\hat{H}$:

1. The empirical entropy estimator, which simply evaluates the Shannon entropy of the empirical distribution of the input sequence. It was shown not to achieve the minimax rates in Shannon entropy estimation [20], and also not to achieve the optimal sample complexity in estimating the entropy rate in Theorem 3 and Corollary 1.

2. The Jiao–Venkat–Han–Weissman (JVHW) estimator, which is based on best polynomial approximation and proved to be minimax rate-optimal in [19]. The independent work [45] is based on similar ideas.

3. The Valiant–Valiant (VV) estimator, which is based on linear programming and proved to achieve the $\frac{S}{\ln S}$ phase transition for Shannon entropy in [43].

4. The profile maximum likelihood estimator (PML), which is proved to achieve the $\frac{S}{\ln S}$ phase transition in [1]. However, there does not exist an efficient algorithm to even approximately compute the PML with provably ganrantees.

There is another estimator, *i.e.*, the Lempel–Ziv (LZ) entropy rate estimator [47], which does not lie in the category of conditional approaches. The LZ estimator estimates the entropy through compression: it is well known that for a universal lossless compression scheme, its codelength per symbol would approach the Shannon entropy rate as length of the sample path grows to infinity. Specifically, for the following random matching length defined by

$$L_i^n = 1 + \max \left\{ 1 \le l \le n : \exists j \le i - 1 \ s.t. \ (X_i, \cdots, X_{i+l-1}) = (X_j, \cdots, X_{j+l-1}) \right\}, \tag{69}$$

it is shown in [46] that for stationary and ergodic Markov chains,

$$\lim_{n \to \infty} \frac{L_i^n}{\ln n} = \bar{H} \ a.s. \tag{70}$$

We use alphabet size $S = 200$ and vary the sample size $n$ from 100 to 300000 to demonstrate how the performance varies as the sample size increases. We compare the performance of the estimators by measuring the root mean square error (RMSE) in the following four different scenarios via 10 Monte Carlo simulations:

1. Uniform: The eigenvalue of the transition matrix is uniformly distributed except the largest one and the transition matrix is generated using the method in [17]. Here we use spectral gap $\gamma = 0.1$.

2. Zipf: The transition probability $T_{ij} \propto \frac{1}{i+j}$.

3. Geometric: The transition probability $T_{ij} \propto 2^{-|i-j|}$.

4. Memoryless: The transition matrix consists of identical rows.

In all of the four cases, the JVHW estimator outperforms the empirical entropy rate. The results of VV [43] and LZ [46] are not included due to their considerable longer running time. For example, when $S = 200$ and $n = 300000$ and we try to estimate the entropy rate from a single trajectory of the Markov chain, the empirical entropy and the JVHW estimator were evaluated in less than 30 seconds. The evaluation of LZ estimator and the conditional VV method did not terminate after a

month.[5] The main reason for the slowness of the VV methods in the context of Markov chains is that for each context it needs to call the original VV entropy estimator (2000 times in total in the above experiment), each of which needs to solve a linear programming.

(a) Uniform

(b) Zipf

(c) Geometric

(d) Memoryless

Figure 2: Comparison of the performances of the empirical entropy rate and JVHW estimator in different parameter configurations

# E   More on fundamental limits of language modeling

Since the number of words in the English language (*i.e.*, our "alphabet" size) is huge, in view of the $\frac{S^2}{\log S}$ result we showed in theory, a natural question is whether a corpus as vast as the 1BW corpus is enough to allow reliable estimates of conditional entropy (as in Figure 1). A quick answer to this question is that our theory has so far focused on the worst-case analysis and, as demonstrated below, natural language data are much nicer so that the sample complexity for accurate estimation is much lower than what the minimax theory predicts. Specifically, we computed the conditional entropy estimates of Figure 1 but this time restricting the sample to only a subset of the corpus. A plot of the resulting estimate as a function of sample size is shown in Figures 3 and 4. Because sentences in the corpus are in randomized order, the subset of the corpus taken is randomly chosen.

To interpret these results, first, note the number of distinct unigrams (*i.e.*, words) in the 1BW corpus is about two million. We recall that in the i.i.d. case, $n \gg S/\ln S$ samples are necessary [41, 45, 19], even in the worst case a dataset of 800 million words will be more than adequate to provide a reliable estimate of entropy for $S \approx 2$ million. Indeed, the plot for unigrams with the JVHW estimator in Figure 3 supports this. In this case, the entropy estimates for all sample sizes greater than 338 000

Figure 3: Estimates of conditional entropy versus sample size for 1BW unigrams; dotted lines are the estimate using the entire corpus (*i.e.*, the final estimate). Note the zoomed-in axes.

Table 1: Convergence points for 1BW conditional entropy estimates (within 0.1 bit of final estimate)

| $k$ | JVHW estimator | | empirical entropy | |
| --- | --- | --- | --- | --- |
| | sample size | % of corpus | sample size | % of corpus |
| 1 | 338k | 0.04% | 2.6M | 0.34% |
| 2 | 77M | 10.0% | 230M | 29.9% |
| 3 | 400M | 54.2% | 550M | 74.5% |

words is within 0.1 bits of the entropy estimate using the entire corpus. That is, it takes just 0.04% of the corpus to reach an estimate within 0.1 bits of the true value.

We note also that the empirical entropy rate converges to the same value, 10.85, within two decimal places. This is also shown in Figure 3. The dotted lines indicate the final entropy estimate (of each estimator) using the entire corpus of $7.7 \times 10^8$ words.

Results for similar experiments with bigrams and trigrams are shown in Figure 4 and Table 2. Since the state space for bigrams and trigrams is much larger, convergence is naturally slower, but it nonetheless appears fast enough that our entropy estimate should be within on the order of 0.1 bits of the true value.

With these observations, we believe that the estimates based on the 1BW corpus should have enough samples to produce reasonably reliable entropy estimates. As one further measure, to approximate the variance of these entropy estimates, we also ran bootstraps for each memory length $k = 1, \ldots, 4$, with a bootstrap size of the same size as the original dataset (sampling with replacement). For the

Table 2: Points at which the 1BW entropy estimates are within 0.1 bit of the final estimate

| $k$ | sample size | % of corpus |
| --- | --- | --- |
| 1 | 338k | 0.04% |
| 2 | 77M | 10.0% |
| 3 | 400M | 54.2% |

Figure 4: Estimates of conditional entropy versus sample size for 1BW bigrams and trigrams; dotted lines are the estimate using the entire corpus (*i.e.*, the final estimate)

Table 3: Bootstrap estimates of error range

| $k$ | PTB estimate | st. dev. | range | 1BW estimate | st. dev. | range |
|---|---|---|---|---|---|---|
| 1 | 10.62 | 0.00360 | 0.0172 | 10.85 | 0.000201 | 0.00091 |
| 2 | 6.68 | 0.00360 | 0.0183 | 7.52 | 0.000152 | 0.00081 |
| 3 | 3.44 | 0.00384 | 0.0159 | 5.52 | 0.000149 | 0.00078 |
| 4 | 1.50 | 0.00251 | 0.0121 | 3.46 | 0.000173 | 0.00081 |

1BW corpus, with 100 bootstraps, the range of estimates (highest less lowest) for each memory length never exceeded 0.001 bit, and the standard deviation of estimates was just 0.0002—that is, the error ranges implied by the bootstraps are too small to show legibly on Figure 1. For the PTB corpus, also with 100 bootstraps, the range never exceeded 0.03 bit. Further details of our bootstrap estimates are given in Table 3.

## F  Auxiliary lemmas

**Lemma 13.** *For an arbitrary sequence* $(x_0, x_1, \ldots, x_n) \in \mathcal{X}^{n+1}, \mathcal{X} = \{1, 2, \ldots, S\}$, *define the empirical distribution of the consecutive pairs as* $\hat{P}_{X_1 X_2} = \frac{1}{n} \sum_{i=0}^{n-1} \delta_{(x_i, x_{i+1})}$. *Let* $\hat{P}_{X_1} = \frac{1}{n} \sum_{i=0}^{n-1} \delta_{x_i}$ *be the marginal distribution of* $\hat{P}_{X_1 X_2}$, *and the empirical frequency of state* $i$ *as*

$$\hat{\pi}_i = \frac{1}{n} \sum_{j=0}^{n-1} \mathbb{1}(x_j = i). \tag{71}$$

*Denote the empirical conditional distribution as* $\hat{P}_{X_2|X_1} = \frac{\hat{P}_{X_1 X_2}}{\hat{P}_{X_1}}$, *i.e.,*

$$\hat{P}_{X_2|X_1=i}(j) = \frac{\sum_{m=1}^{n} \mathbb{1}(x_m = j, x_{m-1} = i)}{n \hat{\pi}_i}, \tag{72}$$

*whenever $\hat{\pi}_i$. Let $\bar{H}_{\text{emp}} = \sum_{i=1}^{S} \hat{\pi}_i H(\hat{P}_{X_2|X_1=i})$ and $H(\cdot)$ is the Shannon entropy. Then, we have*

$$\bar{H}_{\text{emp}} = H(\hat{P}_{X_1 X_2}) - H(\hat{P}_{X_1}), \tag{73}$$

$$= \min_{P \in \mathcal{M}_2(S)} \frac{1}{n} \ln \frac{1}{P_{X_1^n|X_0}(x_1^n|x_0)} \tag{74}$$

*where in (74), for a given transition matrix $P$, $P_{X_1^n|X_0}(x_1^n|x_0) \triangleq \prod_{t=0}^{n-1} P(x_{t+1}, x_t)$.*

The following lemma gives well-known tail bounds for Poisson and Binomial random variables.

**Lemma 14.** *[29, Exercise 4.7] If $X \sim \text{Poi}(\lambda)$ or $X \sim \text{B}(n, \frac{\lambda}{n})$, then for any $\delta > 0$, we have*

$$\mathbb{P}(X \geq (1+\delta)\lambda) \leq \left( \frac{e^\delta}{(1+\delta)^{1+\delta}} \right)^\lambda \leq e^{-\delta^2\lambda/3} \vee e^{-\delta\lambda/3} \tag{75}$$

$$\mathbb{P}(X \leq (1-\delta)\lambda) \leq \left( \frac{e^{-\delta}}{(1-\delta)^{1-\delta}} \right)^\lambda \leq e^{-\delta^2\lambda/2}. \tag{76}$$

The following lemma is the Hoeffding inequality.

**Lemma 15.** *[15] Let $X_1, X_2, \ldots, X_n$ be independent random variables such that $X_i$ takes its value in $[a_i, b_i]$ almost surely for all $i \leq n$. Let $S_n = \sum_{i=1}^{n} X_i$, we have for any $t > 0$,*

$$P\{|S_n - \mathbb{E}[S_n]| \geq t\} \leq 2 \exp\left( -\frac{2t^2}{\sum_{i=1}^{n}(b_i - a_i)^2} \right). \tag{77}$$

# G Proofs of main lemmas

## G.1 Proof of Lemma 4

We being with a lemma on the concentration of the empirical distribution $\hat{\pi}$ for reversible Markov chains.

**Lemma 16.** *Consider a reversible stationary Markov chain with spectral gap $\gamma$. Then, for every $i, 1 \leq i \leq S$, every constant $c_3 > 0$, the event*

$$\mathcal{E}_i = \left\{ |\hat{\pi}_i - \pi_i| \geq c_3 \max\left\{ \frac{\ln n}{n\gamma}, \sqrt{\frac{\pi_i \ln n}{n\gamma}} \right\} \right\} \tag{78}$$

*happens with probability at most $\frac{2}{n^\beta}$, where $\beta = \frac{c_3^2}{4+10c_3}$.*

*Proof of Lemma 16.* Recall the following Bernstein inequality for reversible chains [33, Theorem 3.3]: For any stationary reversible Markov chain with spectral gap $\gamma$,

$$P\left\{ |\hat{\pi}_i - \pi_i| \geq \frac{t}{n} \right\} \leq 2 \exp\left( -\frac{t^2\gamma}{4n\pi_i(1-\pi_i) + 10t} \right). \tag{79}$$

We have $\frac{\ln n}{n\gamma} \geq \sqrt{\frac{\pi_i \ln n}{n\gamma}}$ if and only if $\pi_i \leq \frac{\ln n}{n\gamma}$. We split the proof of (78) into two parts.

1. $\pi_i \leq \frac{\ln n}{n\gamma}$: Invoking (79) and setting $t = c_3 \frac{\ln n}{\gamma}$, we have

$$P_\pi\left( |\hat{\pi}_i - \pi_i| \geq c_3 \frac{\ln n}{n\gamma} \right) \leq 2 \exp\left( -\frac{\gamma c_3^2 \frac{\ln^2 n}{\gamma^2}}{4n \frac{\ln n}{n\gamma} + 10c_3 \frac{\ln n}{\gamma}} \right)$$

$$\leq 2 \exp\left( -\frac{c_3^2}{4 + 10c_3} \ln n \right)$$

$$= \frac{2}{n^\beta}.$$

2. $\pi_i \geq \frac{\ln n}{n\gamma}$: Invoking (79) and setting $t = c_3 \sqrt{\frac{n\pi_i \ln n}{\gamma}}$, we have

$$P_\pi \left( |\hat{\pi}_i - \pi_i| \geq c_3 \sqrt{\frac{\pi_i \ln n}{n\gamma}} \right) \leq 2\exp\left( -\frac{\gamma c_3^2 \frac{n\pi_i \ln n}{\gamma}}{4n\pi_i + 10 c_3 \sqrt{\frac{n\pi_i \ln n}{\gamma}}} \right)$$

$$\leq 2\exp\left( -\frac{c_3^2 n\pi_i \ln n}{4n\pi_i + 10 c_3 n\pi_i} \right)$$

$$= \frac{2}{n^\beta}.$$

$\square$

Now we are ready to prove Lemma 4. We only consider $\mathcal{G}_{\text{opt}}$ and the upper bound on $P(\mathcal{G}_{\text{emp}})$ follows from the same steps. By the union bound, it suffices to upper bound the probability of the complement of each event in the definition of the "good" event $\mathcal{G}_{\text{opt}}$ (cf. Definition 2).

For the first part of the definition, the probability of "bad" events $\mathcal{E}_i^c$ in (25) are upper bounded by

$$\sum_{i \in [S]} P(\mathcal{E}_i^c) \leq S \cdot \frac{2}{n^\beta}, \tag{80}$$

where $\beta = \frac{c_3^2}{4 + 10 c_3}$ as in Lemma 16. Since we have assumed that $c_3 \geq 20$, we have $\beta \geq 1$.

For the second part of the definition, applying Lemma 3, the overall probability of "bad" events $\mathcal{H}_i^c$ in (26) are upper bounded by

$$\sum_{i \in [S]} P(\mathcal{H}_i^c) \mathbb{1}\left( \pi_i \geq n^{c_4-1} \vee 100 c_3^2 \frac{\ln n}{n\gamma} \right)$$

$$\leq \sum_{i=1}^S 2 \cdot c_3 \sqrt{\frac{n\pi_i \ln n}{\gamma}} \frac{2}{\left( n\pi_i - c_3 \sqrt{\frac{n\pi_i \ln n}{\gamma}} \right)^\beta} \mathbb{1}\left( \pi_i \geq n^{c_4-1} \vee 100 c_3^2 \frac{\ln n}{n\gamma} \right)$$

$$\leq \sum_{i=1}^S 2 \cdot c_3 \sqrt{\frac{n\pi_i \ln n}{\gamma}} \frac{2}{(9 n\pi_i / 10)^\beta} \mathbb{1}\left( \pi_i \geq n^{c_4-1} \vee 100 c_3^2 \frac{\ln n}{n\gamma} \right)$$

$$\leq \sum_{i=1}^S 2 c_3 n\pi_i \frac{2}{(9 n\pi_i / 10)^\beta} \mathbb{1}\left( \pi_i \geq n^{c_4-1} \vee 100 c_3^2 \frac{\ln n}{n\gamma} \right)$$

$$= \sum_{i=1}^S \frac{4 c_3 n\pi_i}{(9 n\pi_i / 10)^\beta} \mathbb{1}\left( \pi_i \geq n^{c_4-1} \vee 100 c_3^2 \frac{\ln n}{n\gamma} \right)$$

$$\leq D \frac{S}{n^{c_4(\beta-1)}},$$

where $D \triangleq \frac{4 c_3 (10)^\beta}{9^\beta}$ and the second step follows from the fact that $\pi_i \mapsto n\pi_i - c_3 \sqrt{\frac{n\pi_i \ln n}{\gamma}}$ is increasing when $\pi_i \geq 100 c_3^2 \frac{\ln n}{n\gamma}$.

### G.2  Proof of Lemma 5

We simulate a Markov chain sample path with transition matrix $T_{ij}$ and stationary distribution $\pi_i$ from the independent multinomial model as described in Definition 3, and define the estimator $\hat{H}_2$ as follows: output zero if the event $\cap_{1 \leq i \leq S} \mathcal{E}_i$ does not happen (where $\mathcal{E}_i$ are events defined in Definition 2); otherwise, we set

$$\hat{H}_2(X_0, (W_{ij})_{i \in [S], j \leq m_i}) = \hat{H}_1(X_0, (W_{ij})_{i \in [S], j \leq n_i}).$$

Note that this is a valid definition since $\cap_{1 \leq i \leq S} \mathcal{E}_i$ implies $n_i \leq m_i$ for any $i \in [S]$. As a result,

$$P_{\mathsf{IM}}\left(|\hat{H}_2 - \bar{H}| \geq \epsilon\right) \leq P_{\mathsf{IM}}\left((\cap_{1 \leq i \leq S}\mathcal{E}_i)^c\right) + P_{\mathsf{IM}}\left(\cap_{1 \leq i \leq S}\mathcal{E}_i\right) P_{\mathsf{IM}}\left(|\hat{H}_2 - \bar{H}| \geq \epsilon | \cap_{1 \leq i \leq S} \mathcal{E}_i\right). \tag{81}$$

It follows from Lemma 4 that

$$P_{\mathsf{IM}}\left((\cap_{1 \leq i \leq S}\mathcal{E}_i)^c\right) \leq \frac{2S}{n^\beta}, \tag{82}$$

where $\beta = \frac{c_3^2}{4 + 10c_3} \geq 1$. Now, it suffices to upper bound $P_{\mathsf{IM}}\left(|\hat{H}_2 - \bar{H}| \geq \epsilon | \cap_{1 \leq i \leq S} \mathcal{E}_i\right)$. The crucial observation is that the joint distribution of $(X_0, (n_i)_{i \in [S]}, (W_{ij})_{i \in [S], j \leq n_i})$ are identical in two models, and thus

$$P_{\mathsf{IM}}\left(|\hat{H}_2 - \bar{H}| \geq \epsilon | \cap_{1 \leq i \leq S} \mathcal{E}_i\right) = P_{\mathsf{MC}}\left(|\hat{H}_1 - \bar{H}| \geq \epsilon | \cap_{1 \leq i \leq S} \mathcal{E}_i\right) \tag{83}$$

$$P_{\mathsf{IM}}\left(\cap_{1 \leq i \leq S}\mathcal{E}_i\right) = P_{\mathsf{MC}}\left(\cap_{1 \leq i \leq S}\mathcal{E}_i\right). \tag{84}$$

By definition, the estimator $\hat{H}_1$ satisfies

$$P_{\mathsf{MC}}\left(\cap_{1 \leq i \leq S}\mathcal{E}_i, |\hat{H}_1 - \bar{H}| \geq \epsilon\right) \leq \delta. \tag{85}$$

A combination of the previous inequalities gives

$$P_{\mathsf{IM}}\left(|\hat{H}_2 - \bar{H}| \geq \epsilon\right) \leq \frac{2S}{n^\beta} + \delta. \tag{86}$$

as desired.

### G.3 Proof of Lemma 6

We can simulate the independent multinomial model from the independent Poisson model by conditioning on the row sum. For each $i$, conditioned on $M_i \triangleq \sum_{j=1}^{S} C_{ij} = m_i$, the random vector $C_i = (C_{i1}, C_{i2}, \ldots, C_{iS})$ follows the multinomial distribution $\mathrm{multi}\,(m_i, T_i)$, where $T = T(R)$ is the transition matrix obtained from normalizing $R$. In particular, $T_i = \frac{1}{\sum_{j=1}^{S} R_{ij}}(R_{i1}, R_{i2}, \ldots, R_{iS})$.

Furthermore, $C_1, \ldots, C_S$ are conditionally independent. Thus, to apply the estimator $\hat{H}_1$ designed for the independent multinomial model with parameter $n$ that fulfills the guarantee (51), we need to guarantee that

$$M_i > n\pi_i + c_3 \max\left\{\frac{\ln n}{\gamma^*}, \sqrt{\frac{n\pi_i \ln n}{\gamma^*}}\right\}, \tag{87}$$

for all $i$ with probability at least $1 - Sn^{-c_3/2}$. Here $c_3 \geq 20$ is the constant in Definition 3, and

$$\pi_i = \frac{\sum_{j=1}^{S} R_{ij}}{r}, \tag{88}$$

where $r = \sum_{1 \leq i,j \leq S} R_{ij}$. Note that $M_i \sim \mathrm{Poi}(\lambda_i)$, where $\lambda_i \triangleq \frac{4n}{\tau} \sum_j R_{ij} = \frac{4nr}{\tau} \pi_i \geq 4n\pi_i$, due to the assumption that $r \geq \tau$. By the assumption of $\pi_i \geq \pi_{\min} \geq \frac{c_3 \ln n}{n\gamma}$, we have

$$n\pi_i \geq c_3 \max\left\{\frac{\ln n}{\gamma^*}, \sqrt{\frac{n\pi_i \ln n}{\gamma^*}}\right\}. \tag{89}$$

Then

$$\mathbb{P}\left(M_i < n\pi_i + c_3 \max\left\{\frac{\ln n}{\gamma^*}, \sqrt{\frac{n\pi_i \ln n}{\gamma^*}}\right\}\right) \leq \mathbb{P}\left(\mathrm{Poi}(4n\pi_i) < 2n\pi_i\right)$$

$$\overset{(a)}{\leq} \exp(-n\pi_i/2) \tag{90}$$

$$\overset{(b)}{\leq} n^{-c_3/2}, \tag{91}$$

where (a) follows from Lemma 14; (b) follows from $\pi_i \geq \pi_{\min} \geq \frac{c_3 \ln n}{n\gamma} \geq \frac{c_3 \ln n}{n}$. This completes the proof.

### G.4 Proof of Lemma 10

The dependence diagram for all random variables is as follows:

$$\mathbf{U} \longrightarrow \mathbf{R} \longrightarrow \pi(\mathbf{R}) \longrightarrow X_0$$
$$\downarrow$$
$$\mathbf{C}$$

where $\pi(\mathbf{R}) = (\pi_0(\mathbf{R}), \pi_1(\mathbf{R}), \cdots, \pi_S(\mathbf{R}))$ is the stationary distribution defined in (17) obtained by normalizing the matrix $\mathbf{R}$. Recall that for $i = 1, 2$, $F_i$ denotes the joint distribution on the sufficient statistic $(X_0, \mathbf{C})$ under the prior $\mu_i$. Our goal is to show that $\mathsf{TV}(F_1, F_2) \to 0$. Note that $X_0$ and $\mathbf{C}$ are dependent; however, the key observation is that, by concentration, the distribution of $X_0$ is close to a fixed distribution $P_0$ on the state space $\{0, 1, \cdots, S\}$, where $P_0 \triangleq \frac{1}{S(1+\sqrt{\alpha S})}(S\sqrt{\alpha S}, 1, 1, \cdots, 1)$. Thus, $X_0$ and $\mathbf{C}$ are approximately independent. For clarity, we denote $F_1 = P_{X_0, \mathbf{C}}, F_2 = Q_{X_0, \mathbf{C}}$. By the triangle inequality of the total variation distance, we have

$$\mathsf{TV}(F_1, F_2) \leq \mathsf{TV}(P_{X_0, \mathbf{C}}, P_0 \otimes P_{\mathbf{C}}) + \mathsf{TV}(P_0 \otimes P_{\mathbf{C}}, P_0 \otimes Q_{\mathbf{C}}) + \mathsf{TV}(Q_{X_0, \mathbf{C}}, P_0 \otimes Q_{\mathbf{C}}). \quad (92)$$

To upper bound the first term, note that $\mathbf{C} \to \mathbf{R} \to X_0$ forms a Markov chain. Hence, by the convexity of total variation distance, we have

$$\begin{aligned}
\mathsf{TV}(P_{X_0, \mathbf{C}}, P_0 \otimes P_{\mathbf{C}}) &= \mathbb{E}_{P_{\mathbf{C}}}[\mathsf{TV}(P_{X_0|\mathbf{C}}, P_0)] && (93) \\
&\leq \mathbb{E}_{P_{\mathbf{R}}}[\mathsf{TV}(P_{X_0|\mathbf{R}}, P_0)] \\
&= \mathbb{E}[\mathsf{TV}(\pi(\mathbf{R}), P_0)] \\
&= \frac{1}{2}\left( \mathbb{E}\left|\pi_0(\mathbf{R}) - \frac{1}{1+\sqrt{\alpha S}}\right| + \sum_{i=1}^{S} \mathbb{E}\left|\pi_i(\mathbf{R}) - \frac{1}{S(1+\sqrt{\alpha S})}\right| \right). && (94)
\end{aligned}$$

We start by showing that the row sums of $\mathbf{R}$ concentrate. Let $r_i = \sum_{i=0}^{S} R_{ij} = a + \sum_{i=1}^{S} U_{ij}$, where $a = \sqrt{\alpha S}$. It follows from the Hoeffding inequality in Lemma 15 that

$$\mathbb{P}\left( \left| r_i - (\sqrt{\alpha S} + \alpha S) \right| \geq u, \ i = 1, \ldots, S \right) \leq 2S \exp\left( \frac{-2u^2}{S\left(\frac{d_1^2 S \ln S}{n}\right)^2} \right) \to 0, \quad (95)$$

provided that $u \gg \frac{(S \ln S)^{3/2}}{n}$.

Next consider the entrywise sum of $\mathbf{R}$. Write $r \triangleq \sum_{0 \leq i, j \leq S} R_{ij} = b + 2aS + \sum_{1 \leq i < j \leq S} 2U_{ij} + \sum_{1 \leq i \leq S} U_{ii}$. Note that $\mathbb{E}[r] = b + 2aS + S^2\alpha = S(1+\sqrt{\alpha S})^2$, by (64). Then, it follows from the Hoeffding inequality in Lemma 15 that

$$\mathbb{P}\left( \left| r - S(1+\sqrt{\alpha S})^2 \right| \geq \sqrt{S}u \right) \leq 2 \exp\left( \frac{-2Su^2}{\frac{S(S-1)}{2}4\left(\frac{d_1^2 S \ln S}{n}\right)^2 + S\left(\frac{d_1^2 S \ln S}{n}\right)^2} \right) \to 0 \quad (96)$$

provided that $u \gg \frac{S^{3/2} \ln S}{n}$. Henceforth, we set

$$u = \frac{(S \ln^2 S)^{3/2}}{n}. \quad (97)$$

Hence, with probability tending to one, $|r_i - (\sqrt{\alpha S} + \alpha S)| \leq u$ for $i = 1, 2, \cdots, S$ and $|r - S(1+\sqrt{\alpha S})^2| \leq \sqrt{S}u$. Conditioning on this event, for $i = 1, 2, \cdots, S$ we have

$$\left| \pi_i(\mathbf{R}) - \frac{\sqrt{\alpha S}}{S(1+\sqrt{\alpha S})} \right| \leq (\sqrt{\alpha S} + \alpha S)\left| \frac{1}{r} - \frac{1}{\mathbb{E}[r]} \right| + \frac{u}{r} \leq \frac{2u}{S^{\frac{3}{2}}} + \frac{u}{S}. \quad (98)$$

For $i = 0$, $\pi_0(\mathbf{R}) = \frac{S(1+\sqrt{\alpha S})}{r}$, we have

$$\left| \pi_0(\mathbf{R}) - \frac{1}{1+\sqrt{\alpha S}} \right| = S(1+\sqrt{\alpha S}) \left| \frac{1}{r} - \frac{1}{\mathbb{E}[r]} \right| \leq \frac{2u}{\sqrt{S}}. \tag{99}$$

Therefore, in view of (94), we have

$$\mathsf{TV}(P_{X_0,\mathbf{C}}, P_0 \otimes P_{\mathbf{C}}) \leq \left( \frac{2u}{\sqrt{S}} + \sum_{i=1}^{S} \left( \frac{2u}{S^{\frac{3}{2}}} + \frac{u}{S} \right) \right) + 2 \cdot o(1) = \frac{4u}{\sqrt{S}} + u + o(1) = o(1) \tag{100}$$

as $S \to \infty$. Similarly, we also have $\mathsf{TV}(Q_{X_0,\mathbf{C}}, P_0 \otimes Q_{\mathbf{C}}) = o(1)$.

By (92), it remains to show that $\mathsf{TV}(P_0 \otimes P_{\mathbf{C}}, P_0 \otimes Q_{\mathbf{C}}) = o(1)$. Note that $P_{\mathbf{C}}, Q_{\mathbf{C}}$ are products of Poisson mixtures, by the triangle inequality of total variation distance again we have

$$\mathsf{TV}(P_0 \otimes P_{\mathbf{C}}, P_0 \otimes Q_{\mathbf{C}}) = \mathsf{TV}(P_{\mathbf{C}}, Q_{\mathbf{C}}) \leq \sum_{1 \leq i \leq j \leq S} \mathsf{TV}\left( \mathbb{E}[\mathsf{Poi}(\frac{4n}{\tau}U_{ij})], \mathbb{E}[\mathsf{Poi}(\frac{4n}{\tau}U'_{ij})] \right). \tag{101}$$

We upper bound the individual terms in (101). For the total variation distance between Poisson mixtures, note that the random variables $\frac{4n}{S}U_{ij}$ and $\frac{4n}{S}U'_{ij}$ match moments up to order

$$\frac{D}{\sqrt{\eta}} = Dd_1 \ln S, \tag{102}$$

and are both supported on $[0, \alpha\eta^{-1} \cdot \frac{4n}{S}] = [0, \frac{d_1^2 S \ln S}{n} \frac{4n}{S}] = [0, 4d_1^2 \ln S]$. It follows from Lemma 9 that if

$$Dd_1 \ln S \geq 8e^2 d_1^2 \ln S, \tag{103}$$

we have

$$\mathsf{TV}\left( \mathbb{E}\left[ \mathsf{Poi}\left( \frac{4n}{S}U_{ij} \right) \right], \mathbb{E}\left[ \mathsf{Poi}\left( \frac{4n}{S}U'_{ij} \right) \right] \right) \leq \frac{1}{2^{Dd_1 \ln S}} \tag{104}$$

$$\leq \frac{1}{S^{\frac{D^2 \ln 2}{4e^2}}} \tag{105}$$

$$\leq \frac{1}{S^{100}}. \tag{106}$$

where we set $d_1 = \frac{D}{8e^2}$ and used the fact that $D \geq 100$. By (101),

$$\mathsf{TV}(P_0 \otimes P_{\mathbf{C}}, P_0 \otimes Q_{\mathbf{C}}) \leq S^2 \cdot \frac{1}{S^{100}} = \frac{1}{S^{98}} = o(1) \tag{107}$$

as $S \to \infty$, establishing the desired lemma.

### G.5 Proof of Lemma 11

Let $\Delta = \frac{cS^2}{8n \log S} = \frac{c\alpha S}{8}$, where $c$ is the constant from Lemma 8. Recall that $\phi(x) = x \log \frac{1}{x}$. In view of (65), we have

$$\bar{H}(T(\mathbf{R})) = \frac{1}{r}\left( \sum_{i,j=0}^{S} \phi(R_{ij}) - \sum_{i=0}^{S} \phi(r_i) \right)$$

$$= \frac{1}{r}(\phi(b) + 2S\phi(a) + \phi(b + aS)) + \frac{1}{r}\sum_{i,j=1}^{S} \phi(R_{ij}) - \frac{1}{r}\sum_{i=0}^{S} \phi(r_i)$$

$$= \underbrace{\frac{1}{r}(\phi(b) + 2S\phi(a) + \phi(b + aS))}_{H_1} + \underbrace{\frac{1}{r}\left( 2\sum_{1 \leq i < j \leq S} \phi(U_{ij}) + \sum_{1 \leq i \leq S} \phi(U_{ii}) \right)}_{H_2} - \underbrace{\frac{1}{r}\sum_{i=0}^{S} \phi(r_i)}_{H_3},$$

where the last step follows from the symmetry of the matrix $\mathbf{U}$.

For the first term, note that $|\phi(b) + 2S\phi(a) + \phi(b+aS)| = |\phi(S) + 2S\phi(\sqrt{\alpha}S) + \phi(S(1+\sqrt{\alpha}S))| \leq 10S \log S$. Thus, conditioned on (96), we have

$$\left| \frac{1}{r} - \frac{1}{\mathbb{E}[r]} \right| \leq \frac{u}{S^{\frac{3}{2}}}, \tag{108}$$

where $\mathbb{E}[r] = S(1+\sqrt{\alpha}S)^2$. Put $h_1 \triangleq \frac{\phi(b)+2S\phi(a)+\phi(b+aS)}{S(1+\sqrt{\alpha}S)^2}$, we have

$$|H_1 - h_1| \leq \frac{10u \ln S}{\sqrt{S}}. \tag{109}$$

with probability tending to one.

For the second term, by Definition 4, for any $i, j$, $U_{ij}$ is supported on $[0, \frac{d_1^2 S \ln S}{n}]$. Thus, $\phi(U_{ij})$ is supported on $[0, \frac{d_1^2 S \ln S}{n} \ln \frac{n}{d_1^2 S \ln S}]$ for any $i, j$. Hence, it follows from the Hoeffding inequality in Lemma 15 that

$$\mathbb{P}\left( \left| 2 \sum_{1 \leq i < j \leq S} \phi(U_{ij}) + \sum_{1 \leq i \leq S} \phi(U_{ii}) - S^2 \mathbb{E}[\phi(U)] \right| \geq \frac{\Delta S}{4} \right) \tag{110}$$

$$\leq 2 \exp\left( \frac{-2(\Delta S/4)^2}{\sum_{1 \leq i < j \leq S} \left( 2\frac{d_1^2 S \ln S}{n} \ln \frac{n}{d_1^2 S \ln S} \right)^2 + \sum_{1 \leq i \leq S} \left( \frac{d_1^2 S \ln S}{n} \ln \frac{n}{d_1^2 S \ln S} \right)^2} \right) \tag{111}$$

$$\leq 2 \exp\left( -\Omega\left( \frac{S^2}{(\ln n)^2 (\ln S)^4} \right) \right) \tag{112}$$

$$\to 0 \tag{113}$$

as $S \to \infty$, provided that $\ln n \ll \frac{S}{\ln^2 S}$. Put $h_2 = \frac{S}{(1+\sqrt{\alpha}S)^2} \mathbb{E}[\phi(U)]$. Using (108) and the fact that $0 \leq \phi(U) \leq \frac{d_1^2 S \ln S}{n} \ln \frac{n}{d_1^2 S \ln S}$, we have

$$|H_2 - h_2| \leq u \cdot \frac{d_1^2 S^{\frac{3}{2}} \ln S}{n} \ln \frac{n}{d_1^2 S \ln S} + \frac{\Delta}{4}. \tag{114}$$

For the third term, condition on the event in (95), we have $|\phi(r_i) - \phi(\alpha S + \sqrt{\alpha}S)| \leq Cu \ln S$ and $|\phi(r_i)| \leq C$, for some absolute constant $C$. Put $h_3 = \frac{1}{(1+\sqrt{\alpha}S)^2} \phi(\alpha S + \sqrt{\alpha}S)$. We have

$$|H_3 - h_3| \leq \frac{Cu}{\sqrt{S}} + Cu \ln S. \tag{115}$$

Finally, combining (109), (114), (115) as well as (97), with probability tending to one,

$$|\bar{H}(T(\mathbf{R})) - (h_1 + h_2 - h_3)| \leq \frac{\Delta}{4} + C' \frac{S^{3/2} \ln^4 S}{n} \tag{116}$$

for some absolute constant $C'$. Likewise, with probability tending to one, we have

$$|\bar{H}(T(\mathbf{R}')) - (h_1 + h_2' - h_3)| \leq \frac{\Delta}{4} + C' \frac{S^{3/2} \ln^4 S}{n} \tag{117}$$

where $h_2' = \frac{S}{(1+\sqrt{\alpha}S)^2} \mathbb{E}[\phi(U')]$. In view of Lemma 8, we have

$$|h_2 - h_2'| \geq \frac{c\alpha S}{(1 + \sqrt{\alpha}S)^2} \geq 2\Delta. \tag{118}$$

This completes the proof.

### G.6 Proof of Lemma 12

We only consider the random matrix $\mathbf{R} = (R_{ij})$ which is distributed according to the prior $\mu_1$; the case of $\mu_2$ is entirely analogous.

First we lower bound $\pi_{\min}$ with high probability. Recall the definition of $u$ in (97), and (95), (96). Since $u \leq \alpha S = \frac{S^2}{n \ln S}$, we have $S \leq r \leq 5S$ and $r_i \geq \sqrt{\alpha S}$ for all $i \in [S]$ with probability tending to one. Furthermore, $r_0 = S(1 + \sqrt{\alpha S}) \geq \sqrt{\alpha S}$. Consequently,

$$\pi_{\min} = \frac{1}{r} \min_{0 \leq i \leq S} r_i \geq \frac{\sqrt{\alpha S}}{5S} = \frac{1}{5\sqrt{n \ln S}},$$

as desired.

Next, we deal with the spectral gap. Recall $T = T(R)$ is the normalized version of $R$. Let $D = \text{diag}(r_0, \ldots, r_S)$ and $D_\pi = \text{diag}(\pi_0, \ldots, \pi_S)$, where $r_i = \sum_{j=0}^{S} R_{ij}$, $r = \sum_{i,j=0}^{S} R_{ij}$, and $\pi_i = \frac{r_i}{r}$. Then we have $T = D^{-1}R$. Furthermore, by the reversiblity of $T$,

$$T' \triangleq D_\pi^{1/2} T D_\pi^{-1/2} = D^{-1/2} R D^{-1/2}, \tag{119}$$

is a symmetric matrix. Since $T'$ is a similarity transform of $T$, they share the same spectrum. Let $1 = \lambda_1(T) \geq \ldots \geq \lambda_{S+1}(T)$ (recall that $T$ is an $(S+1) \times (S+1)$ matrix). In view of (63), we have

$$L \triangleq \mathbb{E}[\mathbf{R}] = \begin{bmatrix} b & a \cdots a \\ \hline a & \alpha \cdots \alpha \\ \vdots & \vdots \\ a & \alpha \cdots \alpha \end{bmatrix}, \quad Z \triangleq \mathbf{R} - \mathbb{E}[\mathbf{R}] = \begin{bmatrix} 0 & 0 \cdots 0 \\ \hline 0 & \\ \vdots & \mathbf{U} - \mathbb{E}[\mathbf{U}] \\ 0 & \end{bmatrix}$$

Crucially, the choice of $a = \sqrt{\alpha S}, b = S$ in (64) is such that $b\alpha = a^2$, so that $\mathbb{E}[\mathbf{R}]$ is a symmetric positive semidefinite rank-one matrix. Thus, we have from (119)

$$T' = D^{-1/2} L D^{-1/2} + D^{-1/2} Z D^{-1/2}.$$

Note that $L' \triangleq D^{-1/2} L D^{-1/2}$ is also a symmetric positive semidefinite rank-one matrix. Let $\lambda_1(L') \geq 0 = \lambda_2(L') = \cdots = \lambda_{S+1}(L')$. By Weyl's inequality [38, Eq. (1.64)], for $i = 2, \ldots, S+1$, we have

$$|\lambda_i(T)| \leq \|D^{-1/2} Z D^{-1/2}\|_2 \leq \|D^{-1/2}\|_2^2 \|Z\|_2 = \frac{1}{\min_{0 \leq i \leq S} r_i} \|\mathbf{U} - \mathbb{E}[\mathbf{U}]\|_2. \tag{120}$$

Here and below $\| \cdot \|_2$ stands for the spectral norm (largest singular values). So far everything has been determinimistic. Next we show that with high probability, the RHS of (120) is at most $\Omega(\sqrt{\frac{S \ln^3 S}{n}})$.

Note that $\mathbf{U} - \mathbb{E}[\mathbf{U}]$ is a zero-mean Wigner matrix. Furthermore, $U_{ij}$ takes values in $[0, \alpha\eta^{-1}] = [0, \frac{d_1^2 S \ln S}{n}]$, where $d_1$ is an absolute constant. It follows from the standard tail estimate of the spectral norm for the Wigner ensemble (see, e.g. [38, Corollary 2.3.6]) that there exist universal constants $C, c, c' > 0$ such that

$$P\left(\|\mathbf{U} - \mathbb{E}[\mathbf{U}]\|_2 > \frac{C S^{3/2} \ln S}{n}\right) \leq c' e^{-cS}. \tag{121}$$

Combining (95), (120), and (121), the absolute spectral gap of $T = T(\mathbf{R})$ satisfies

$$\mathbb{P}\left(\gamma^*(T(\mathbf{R})) \geq 1 - C\sqrt{\frac{S \ln^3 S}{n}}\right) \to 1,$$

as $S \to \infty$. By union bound, we have shown that $\mathbb{P}(\mathbf{R} \in \mathcal{R}(S, \gamma, \tau, q)) \to 1$, with $\gamma, \tau, q$ as chosen in Lemma 12.

### G.7 Proof of Lemma 13

The representation (73) follows from definition of conditional entropy. It remains to show (74). Let $\hat{P}$ denote the transition matrix corresponding to the empirical conditional distribution, that is, $\hat{P}_{ij} \triangleq \hat{P}_{X_2=j|X_1=i}$. Then, for any transition matrix $P = (P_{ij})$,

$$
\begin{aligned}
\frac{1}{n} \ln \frac{1}{P_{X_1^n|X_0}(x_1^n|x_0)} &= \frac{1}{n} \sum_{m=1}^{n} \sum_{i=1}^{S} \sum_{j=1}^{S} \mathbb{1}(x_{m-1}=i, x_m=j) \ln \frac{1}{P_{ij}} \\
&= \frac{1}{n} \sum_{i=1}^{S} \sum_{j=1}^{S} (n\hat{\pi}_i \hat{P}_{ij}) \ln \frac{1}{P_{ij}} \\
&= \sum_{i=1}^{S} \hat{\pi}_i \sum_{j=1}^{S} \hat{P}_{ij} \ln \frac{1}{P_{ij}} \\
&= \bar{H}_{\mathsf{emp}} + \sum_{i=1}^{S} \hat{\pi}_i D(\hat{P}_{i\cdot} \| P_{i\cdot}),
\end{aligned}
$$

where in the last step $D(p\|q) = \sum_i p_i \ln \frac{p_i}{q_i} \geq 0$ stands for the Kullback–Leibler (KL) divergence between probability vectors $p$ and $q$. Then (74) follows from the fact that the nonnegativity of the KL divergence.

## Footnotes

[4]Note that effectively we are taking a union over the value of $n_i$ instead of conditioning. In fact, conditioned on $n_i = m$, $W_{i1}, \ldots, W_{im}$ are no longer i.i.d. as $T_i$.

[5]For LZ, we use the Matlab implementation in https://www.mathworks.com/matlabcentral/ fileexchange/51042-entropy-estimator-based-on-the-lempel-zivalgorithm?focused= 3881655&tab=function. For VV, we use the Matlab implementation in http://theory.stanford.edu/ ~valiant/code.html. We use 10 cores of a server with CPU frequency 1.9GHz.