[Reviews · NeurIPS 2018]

Reviewer 1



The paper proposes an entropy estimate for Markov chains by reduction to optimal entropy estimation for i.i.d samples. Sample complexity analysis is provided for different mixing scenarios with a minimax rate established for a particular rate. The estimator is used to assess the capacity of language models. This is a very clear and well-written paper. I appreciate the efforts done by the authors to summarize the results. My main question, however, is regarding the experiments. Can the authors comment on the confidence of the entropy estimate given the large size of the alphabet? This is crucial to justify its use for the suggested application. Also, could the authors include the empirical entropy estimates? ============================================================= UPDATE: I thank the authors for the their response. I believe the empirical entropy results setrengthen the paper and I recommend including them in the final version.

Reviewer 2



This paper studies estimation of the Shanon entropy for discrete Markov chains on finite state space of cardinal S, which sufficiently mix. The authors first give two kind of estimators for the Shanon entropy which are consistent in the asymptotic S \to \infty. Then they provide positive lower bound on the bias of the first one if the number of samples is of order S^2, and a minimax result for the second estimator. In a last section, the authors give some applications of their result to language modeling. More specifically, they estimate entropy rate of English based on two datasets to be able to discuss on efficacy of existing models. Overall, I think it is a very nice contribution, well written. Authors' comments on their result are good. Finally, the numerical application of their results is interesting. The only suggestion I could make is that the author could illustrate their theoretical results by some numerical experiments.

Reviewer 3



The paper proposes a new way to estimate entropy from Markov Chain samples and provides an analysis of how many samples are enough for consistent estimation. I can’t say that I understand the paper well enough to do a good judgement, but the results seem interesting and useful, and I like the example experiment. Can you please shed some light on why it could be the case that your theory suggests that PTB should be easier than 1BW, but in practice opposite holds true (Fig 1)?